# Tuning of lattice oxygen reactivity and scaling relation to construct better oxygen evolution electrocatalyst

Zhen-Feng Huang[1,2,8], Shibo Xi[3,8], Jiajia Song[4,5], Shuo Dou[1], Xiaogang Li[1], Yonghua Du [3,7], Caozheng Diao [6], Zhichuan J. Xu [4] & Xin Wang [1✉]

Developing efficient and low-cost electrocatalysts for oxygen evolution reaction is crucial in realizing practical energy systems for sustainable fuel production and energy storage from renewable energy sources. However, the inherent linear scaling relation for most catalytic materials imposes a theoretical overpotential ceiling, limiting the development of efficient electrocatalysts. Herein, using modeled $Na_xMn_3O_7$ materials, we report an effective strategy to construct better oxygen evolution electrocatalyst through tuning both lattice oxygen reactivity and scaling relation via alkali metal ion mediation. Specifically, the number of $Na^+$ is linked with lattice oxygen reactivity, which is determined by the number of oxygen hole in oxygen lone-pair states formed by native Mn vacancies, governing the barrier symmetry between O–H bond cleavage and O–O bond formation. On the other hand, the presence of $Na^+$ could have specific noncovalent interaction with pendant oxygen in *OOH to overcome the limitation from linear scaling relation, reducing the overpotential ceiling. Combining in situ spectroscopy-based characterization with first-principles calculations, we demonstrate that an intermediate level of $Na^+$ mediation ($NaMn_3O_7$) exhibits the optimum oxygen evolution activity. This work provides a new rational recipe to develop highly efficient catalyst towards water oxidation or other oxidative reactions through tuning lattice oxygen reactivity and scaling relation.

[1] School of Chemical and Biomedical Engineering, Nanyang Technological University, Singapore, Singapore. [2] Key Laboratory for Green Chemical Technology of the Ministry of Education, School of Chemical Engineering and Technology, Tianjin University, Tianjin, China. [3] Institute of Chemical and Engineering Sciences, A*STAR, Jurong Island, Singapore. [4] School of Materials Science and Engineering, Nanyang Technological University, Singapore, Singapore. [5] Institute of Molecular Aggregation Science, Tianjin University, Tianjin, PR China. [6] Singapore Synchrotron Light Source, National University of Singapore, Singapore, Singapore. [7] Present address: National Synchrotron Light Source II, Brookhaven National Laboratory, Upton, NY, USA. [8] These authors contributed equally: Zhen-Feng Huang, Shibo Xi. ✉email: WangXin@ntu.edu.sg

The oxygen evolution reaction (OER) is a key reaction and constitutes the bottleneck in many energy conversion and storage systems such as water electrolyzers, rechargeable metal-air batteries and regenerative fuel cells[1–3], due to its intrinsically sluggish kinetics[2,4–6]. Considering the origin of $O_2$ product, there are two widely accepted OER mechanisms including adsorbate evolution mechanism and lattice oxygen oxidation mechanism[5,7]. Regardless of which OER mechanism is applicable on a catalyst surface, it has been reported that O–O bond formation can follow two different pathways, i.e., acid-base nucleophilic attack and O–O direct coupling[7,8]. For the former, there is an inherent linear scaling relation (LSR) between the adsorption energy of *OOH and *OH intermediates. One implication of the above LSR is that the key steps of O–H bond cleavage and *OOH formation are mutually competing, rendering a minimum theoretical overpotential of ~0.4 eV even for the best possible material[5,6]. For the latter, it is not subject to such LSR constraint, but the specific catalytic structural motif to trigger O–O direct coupling is difficult to realize for most catalytic materials[7–9]. Therefore, current research efforts are mainly directed to optimize the pathway of acid-base nucleophilic attack and overcome the limitation from such LSR for developing practical electrocatalysts.

Activating lattice oxygen to generate spin-characteristic ligand holes can tune the lattice oxygen reactivity that links to energy barrier symmetry between O–H bond cleavage and *OOH formation. Further reducing overpotential ceiling requires selective stabilization of *OOH over *OH to overcome such LSR[5,10,11]. Prior studies have indicated that the introduction of hydrated alkali metal ions ($A^+$) as promoter can stabilize the key intermediates or transition states via the noncovalent interaction[12–14]. Inspired by this, $A_xMn_3O_7$ ($0 < x \leq 2$) materials, as one type of alkali metal-incorporated metal oxides, can be a good platform for unveiling how to rationally design better OER electrocatalysts through tuning lattice oxygen reactivity and scaling relation mediated by alkali metal ion. On one hand, the native Mn vacancies in $MnO_2$ layers generate oxygen lone-pair states ($|O_{2p}$), which provides a necessary condition for activating lattice oxygen in view of structural stability[15,16]. On the other hand, the alkali metal ions are directly incorporated into the $MnO_2$ interlayers, offering the noncovalent interaction between alkali metal and *OOH, and a reduced theoretical overpotential ceiling can be expected. Additionally, the Mn migration would be suppressed if the ionic radii of A and Mn differ largely[17]. In this regard, $Na^+$ (102 pm) shows larger contrast in ionic radii with high-valence $Mn^{4+}$ (54 pm) and its use in the compound looks promising.

Herein, we use $Na_xMn_3O_7$ with tunable number of $Na^+$ as model to unlock the specific coordination configuration that can regulate the barrier symmetry between O–H bond cleavage and *OOH formation on the basis of overcoming the LSR between *OOH and *OH. Combining theoretical and experimental approaches, we reveal that the number of $Na^+$ is critical to the overall activity improvement. In terms of electronic effect, the O–O bond formation is promoted as the number of $Na^+$ reduces, because of the increased number of oxygen holes in $|O_{2p}$ upon activating lattice oxygen. Correspondingly, the relative barrier between O–H bond cleavage and O–O bond formation is regulated. Contrarily, in terms of geometric effect, the overpotential ceiling increases as the number of $Na^+$ reduces, because of the weakening of $Na^+$-specific stabilizing effect on pendant oxygen in *OOH. As a result of the above two opposite effects, an intermediate level of $Na^+$ mediation, in this case, $NaMn_3O_7$, exhibits the optimum OER activity. This work provides a guideline for the development of better catalysts towards water oxidation or other oxidative reactions through tuning both lattice oxygen reactivity and scaling relation.

## Result

**Pathway competition for O–O bond formation.** Using density functional theory (DFT) calculations, we firstly probe the regions of space where oxygen lone-pair states locate and then unraveled how spin-characteristic ligand holes is generated upon activating lattice oxygen (Fig. 1 and Supplementary Figs. 1–3). For modeled $Na_xMn_3O_7$ ($Na_{2x/7}(Mn_{6/7}\square_{1/7})O_2$, $\square$ represents Mn vacancy, $x$ = 2, 1.5, 1, and 0.5) slabs, there are two kinds of oxygen ions (Fig. 1a and Supplementary Fig. 3), in which O1 is coordinated with three Mn ions and O2 is coordinated with two Mn ions, respectively. As such, one of the O(2p) orbitals pointing toward Mn vacancy in O2 coordination environment is non-bonded. According to the amplitude of charge transfer energy ($\Delta$) and $d$–$d$ Coulomb interaction ($U$)[18], $Na_2Mn_3O_7$ ($U > \Delta$) is located at charge-transfer regime, showing an empty metallic band lying above the fully filled $|O_{2p}$ band (Fig. 1a), which is evidenced by the projected DOS of the Mn(3d) and O(2p) orbitals (shaded region around $E_F$ with the dominant oxygen character in Fig. 1b). As confirmed from the projected density of states and partial charge density near $E_F$ (Fig. 1b, c and Supplementary Table 1), more oxygen holes from $|O_{2p}$ are generated upon activating lattice oxygen as the number of $Na^+$ reduces. Moreover, such generated oxygen holes are stabilized through π-type interaction between the occupied O 2p and the occupied Mn-$t_{2g}$ in $Mn_{oct}O_6$ where Mn would be in their maximum achievable oxidation states of 4+[19,20]. More specifically, we investigated six representative O2 coordination environments (from S1 to S6 based on the number in neighboring $Na^+$, Fig. 1c) for $Na_xMn_3O_7$ to quantify the lattice oxygen reactivity using the number of oxygen holes ($h^O$). As seen, the magnetization moments of these oxygen ions increased from 0.11 $\mu_B$ (S1) to 0.60 $\mu_B$ (S6), which is further confirmed by Bader charge analysis as the charges of such oxygen ions increase from $-1.03$ e (S1) to $-0.66$ e (S6). More details about calculating the number of oxygen hole via crystal field theory, magnetization moment, and bader charge can be found in Supplementary Fig. 1.

Considering the dual possible roles of oxygen holes in O–O bond formation via either acid-base nucleophilic attack or O–O direct coupling, we further unravel the pathway competition from the perspectives of both thermodynamics and kinetics. In detail, we systematically correlate both pathways with the O2 coordination environments by comparing the relative stabilities and activation free energy barriers ($\Delta G^\ddagger$) between the corresponding isomeric intermediates of A3 and R4 (Fig. 1d and Supplementary Figs. 4–6)[8,21,22]. Based on our previous work[8], the most possible coordination environments to trigger O–O direct coupling are S3, S4 and S6. Therefore, these three environments were investigated. The intermediate of A3 on the former two environments are 1.20 and 0.87 eV/intermediate lower in energy than R4, respectively (Fig. 1d). For S6, A3 is 0.53 eV/intermediate higher in energy than R4 due to the increased numbers of oxygen hole, however, the $\Delta G^\ddagger$ for direct O–O coupling is as high as 1.20 eV (Fig. 1e and Supplementary Fig. 7). All these calculations suggest that the acid-base nucleophilic attack prevails owing to the high energy penalty for breaking the directional Mn-O bond for the O–O direct coupling. Moreover, because the ligand holes provide the spin-controlled electron transfer channels between catalyst and reactant[23], the $\Delta G_{*OOH}$ for *OOH formation is lowered gradually with the greater number of oxygen holes (Fig. 1f).

**Tuning of lattice oxygen reactivity and scaling relation via alkali metal mediation.** The number of $Na^+$ provides the lever to optimize the overall activity through tuning the lattice oxygen reactivity and scaling relation (Fig. 2a and Supplementary Fig. 8). To correlate the change of barrier symmetry resulted from oxygen

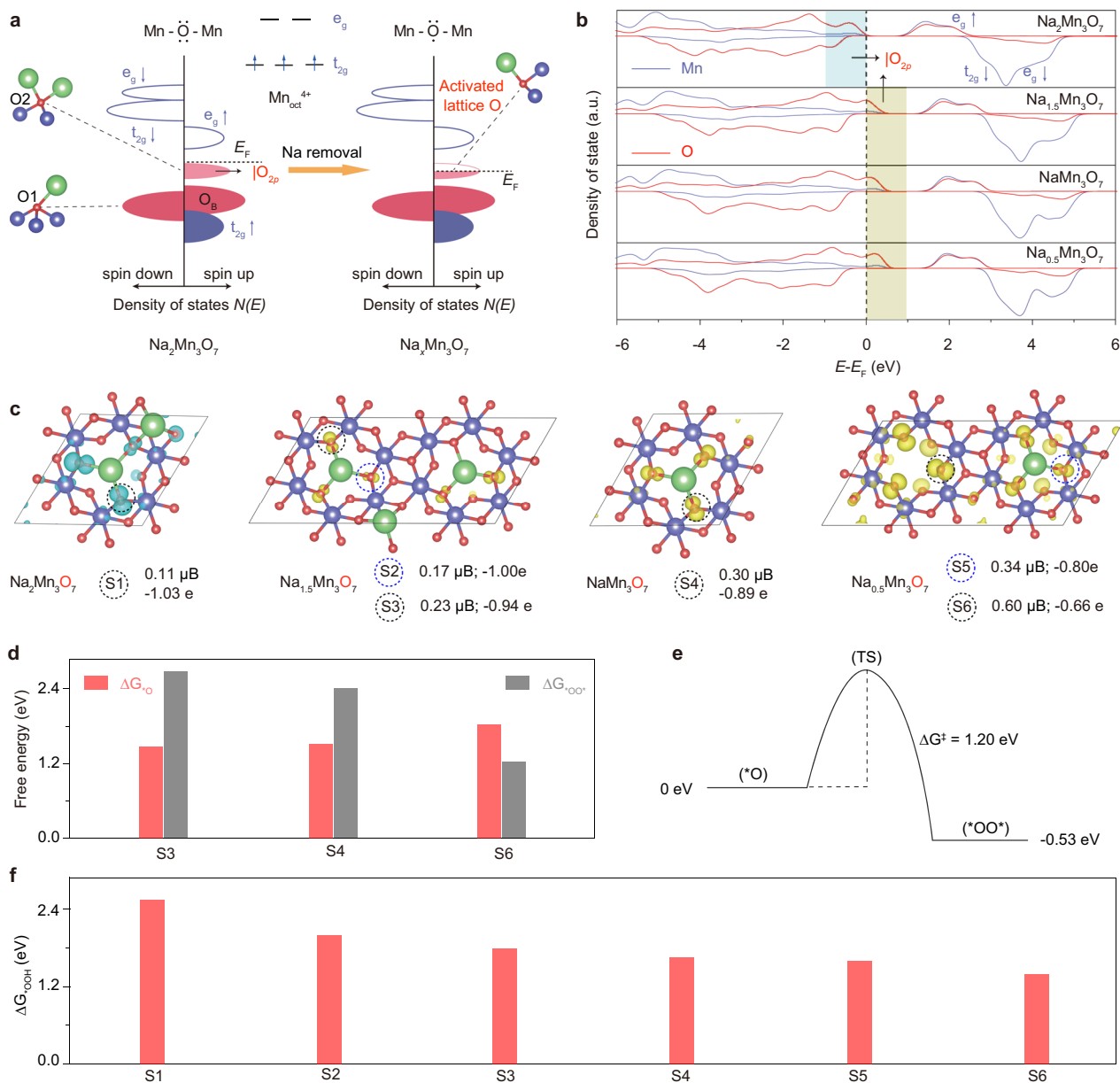

**Fig. 1 Activating lattice oxygen to regulate the pathway competition for O–O bond formation. a** Schematic formation of oxygen holes in $|O_{2p}$ lone-pair states for $Na_xMn_3O_7$. Lattice oxygen atoms are defined as O1 and O2, where O1 is coordinated with three Mn ions and O2 is coordinated with two Mn ions, respectively. **b** Projected density of states of $Na_xMn_3O_7$ slabs ($x = 2, 1.5, 1$, and 0.5). **c** Partial charge density projected on O atoms by the shaded region shown in Fig. 1b. **d** Free energy difference between the two isomeric intermediates of A3 (*O) and R4 (*OO*) on representative oxygen coordination environments. The details about A3 and R4 are shown in Supplementary Fig. 4. **e** Activation free energy barrier ($\Delta G^{\ddagger}$) from A3 to R4 on the specific coordination environment of S6. **f** Free energy barrier ($\Delta G_{*OOH}$) for the formation of *OOH.

hole[24], the electronic parameter (P1) is defined as the variation of $\Delta G_{*O} - \Delta G_{*OH}$ with reference to S1. As seen, the greater number of oxygen holes, the higher value of P1 (Fig. 2b and Supplementary Fig. 9a and Tables 2, 3). Due to the strong LSR of $\Delta G_{*OOH} = \Delta G_{*OH} + 3.24$ eV, the overpotential ceiling of ~0.39 eV is ultimately approached when the magnetization moment (charge) of oxygen ions increases above 0.23 μB (−0.94 e). To demonstrate $Na^+$-specific noncovalent interaction with *OOH[24], the geometric parameter (P2) is then defined as the variation of $\Delta G_{*OOH}$ with reference to S6. As seen, the greater number of $Na^+$ around lattice oxygen, the lower value of P2 (Fig. 2c and Supplementary Tables 2, 4). Owing to the regulation of the LSR, the overpotential ceiling is gradually reduced from 0.39 to 0.19 V. The decreased intersection angles between O–O and H in *OOH

further confirm the enhanced electrostatic interaction between *OOH and $Na^+$ (Supplementary Fig. 9b).

We further build a dynamic volcano plot to decipher the optimal coordination environment for the lowest OER overpotential (Fig. 2d and Supplementary Fig. 10 and Tables 2–5)[11,25,26]. As shown in Supplementary Tables 3 and 4, the individual P1 or P2 optimization ([0, P2] or [P1, 0]) cannot guarantee to achieve low overpotential. This is because what defines an ideal OER catalyst is not that the $\Delta G_{*OOH} - \Delta G_{*OH}$ be 2.46 eV or $\Delta G_{*O} - \Delta G_{*OH}$ be 1.23 eV but rather that all of free energy change of OER steps are numerically equal to the equilibrium potential[10]. Various overpotentials of 0.37, 0.45, 0.33, and 0.41 eV can be achieved in the coordination environments from S2 to S5 (Fig. 2d) with the variation of both P1 and P2. All

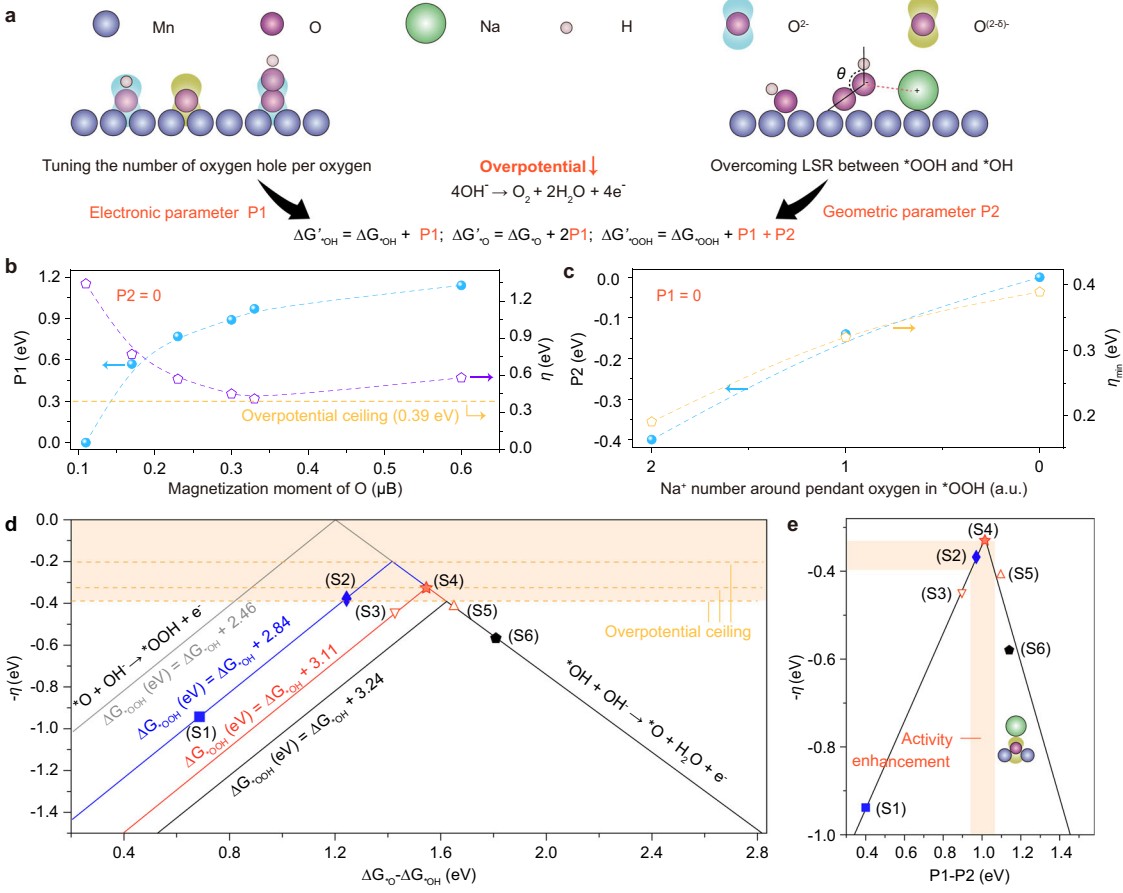

**Fig. 2 Constructing better OER electrocatalyst through tuning the lattice oxygen reactivity and scaling relation. a** Scheme of rational design of better $Na_xMn_3O_7$ electrocatalysts. **b** Shifts in P1 to regulate the theoretical overpotential ($\eta$) by tuning the magnetization moment of O. **c** Shifts in P2 to reduce the overpotential ceiling ($\eta_{min}$) by tuning $Na^+$-specific noncovalent interaction to overcome the LSR. **d** Dynamic volcano plot ($\eta$ versus $\Delta G_{*O} - \Delta G_{*OH}$) derived from the rebuilt LSR. **e** Unified volcano plot ($\eta$ versus P1–P2) using a binary descriptor of P1–P2.

these suggest that tuning of lattice oxygen reactivity and scaling relation via alkali metal mediation can construct better oxygen evolution electrocatalyst. For a better illustration of the overall activity trend, a binary descriptor of P1–P2 that takes the barrier symmetry between the O–H bond cleavage and *OOH formation and overcoming the limitation from LSR into consideration, is further proposed. As shown in Fig. 2e, the overpotentials on different oxygen coordination environments in $Na_xMn_3O_7$ are plotted as a volcano-like function of the calculated P1-P2. More specifically, S4 is closest to the apex and thus considered as the most reactive oxygen site for OER. On the left branch, the OER activity is constrained by lattice oxygen reactivity with the rate-limiting step of *OOH formation; on the right branch, the OER activity is constrained by the scaling relation with the rate-limiting step of O–H bond cleavage. All these demonstrate that by tuning the lattice oxygen reactivity and scaling relation, $NaMn_3O_7$ is predicted to be the optimal electrocatalyst in the modeled $Na_xMn_3O_7$ materials.

**Synthesis and characterization of $Na_xMn_3O_7$.** $Na_xMn_3O_7$ ($x = 2.0, 1.5, 1.0$, and $0.7$) materials with different geometric and electronic environments of lattice oxygen are obtained using a solid-state reaction (see "Methods" and Supplementary Information). XRD pattern of $Na_2Mn_3O_7$ (Fig. 3a) demonstrates that the as-synthesized material is triclinic P1-phase $Na_2Mn_3O_7$ without the presence of impurity phase[17,27]. XRD patterns of other materials (Supplementary Fig. 11) are similar to $Na_2Mn_3O_7$ with no impurity peak, which is ascribed to the structural

flexibility due to the presence of native Mn vacancy in $MnO_2$ layer[27]. Furthermore, the diffraction peaks (16.3°, 32.5°, and 38.8°) exhibit a little shift to a higher angle, suggesting the gradual lattice contraction with the decreased number of $Na^+$. The Na/Mn ratios obtained by inductively coupled plasma optical emission spectrometry (ICP-MS) are roughly close to the starting materials (Supplementary Table 6). SEM and transmission electron microscopy (TEM) images combined with energy dispersive spectrometer (EDS) mapping further demonstrate the layered structure of the samples with a homogenous distribution of Na, Mn, and O elements (Supplementary Figs. 12–14).

We further investigated the electronic and coordination structures of $Na_xMn_3O_7$ from bulk to surface using hard and soft X-ray absorption spectroscopy (XAS). As shown in X-ray absorption near edge structure (XANES) spectra, the energy of Mn K-edge for $Na_2Mn_3O_7$ is very close to that of the referenced $MnO_2$ (Fig. 3b), indicating it predominantly consists of $Mn^{4+}$ ions. As the number of $Na^+$ decreases, the Mn K-edge shifts to a lower energy region and the intensity of white line peak is decreased. This indicates that Mn oxidation state in $Na_xMn_3O_7$ is lowered with the presence of oxygen vacancy as the number of $Na^+$ decreases[28]. In addition, the almost overlapped Mn K-edge oscillation curves (Fig. 3c) indicate their similar geometric structures, in accordance with XRD patterns (Supplementary Fig. 11). Fourier-transformed magnitude plot of the Mn K-edge extended X-ray absorption fine structure (EXAFS) spectra are shown in Fig. 3d, where the two shells at 1.2 and 2.5 Å represent the Mn–O scattering path and Mn–Mn scattering path[29],

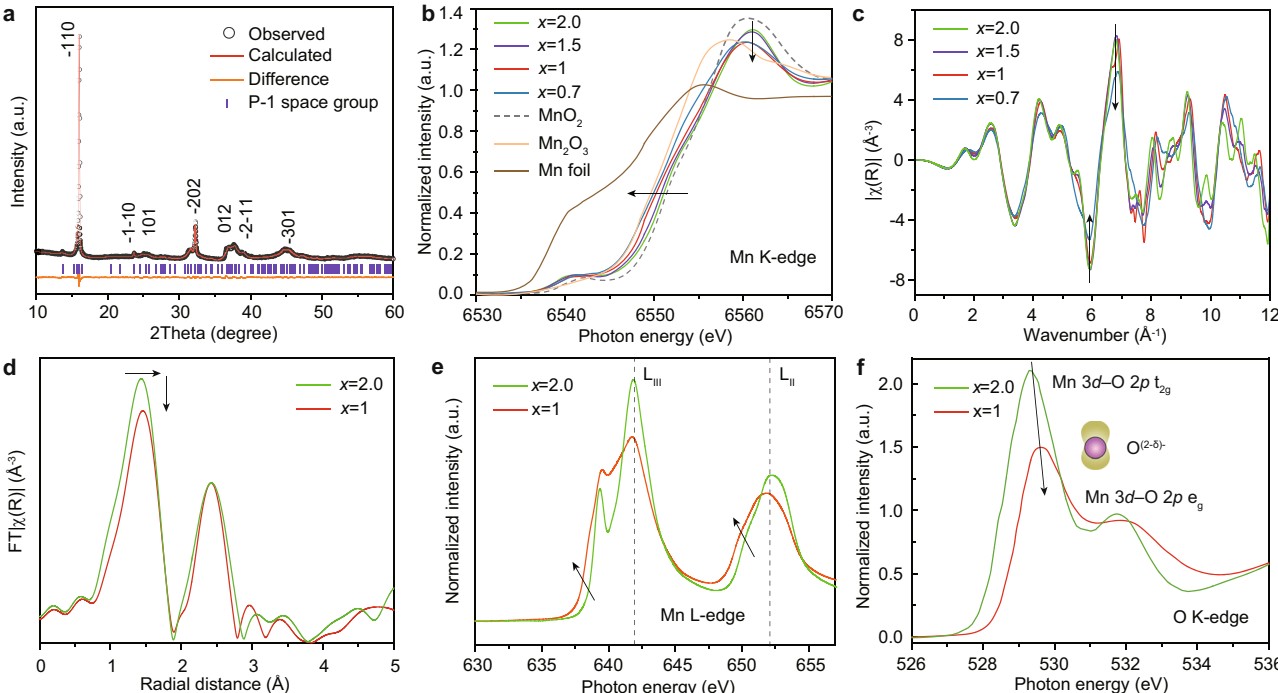

**Fig. 3 Electronic and atomic coordination structures of Na$_x$Mn$_3$O$_7$. a** XRD pattern of Na$_2$Mn$_3$O$_7$. Normalized Mn K-edge XANES spectra (**b**) and EXAFS oscillation functions (**c**) of Na$_x$Mn$_3$O$_7$ ($x$ = 2.0, 1.5, 1.0 and 0.7). **d** Fourier transform magnitudes of Mn K-edge EXAFS $k^2\chi(k)$. Mn L$_{2,3}$-edge (**e**) and O K pre-edge (**f**) of soft XAS spectra of Na$_2$Mn$_3$O$_7$ and NaMn$_3$O$_7$.

respectively. As the number of Na$^+$ is reduced, the coordination number of Mn–O decreases while the interatomic distance of Mn–O increases (Supplementary Table 7), further suggesting the presence of oxygen vacancy and Mn$^{3+}$ ions in the bulk[30]. Meanwhile, the negligible changes of Mn–Mn peak imply unchanged Mn–Mn interatomic distance and coordination number, reflecting the strong structural flexibility to accommodate distortion on transition metals from the native Mn vacancy in MnO$_2$ layer[17,27]. Soft X-ray absorption spectroscopy that is more surface sensitive, were further performed on Na$_2$Mn$_3$O$_7$ and NaMn$_3$O$_7$. From the Mn L-edge spectra (Fig. 3e), the two main peaks at the L$_{III}$ and L$_{II}$ edges, which arises from the transition of a 2$p$ electron to the partially filled 3$d$ shell[31], are significantly changed in shape (less sharp e$_g$/t$_{2g}$ feature) and position (toward lower energy direction), suggesting the increased delocalized feature of electronic state and decreased oxidation state of Mn ions for NaMn$_3$O$_7$ compared with Na$_2$Mn$_3$O$_7$. From the O K pre-edge spectra (Fig. 3f), the characteristic peaks between 528 and 534 eV represent the spectroscopic excitations to the hybridized state of O-2$p$ and Mn-3$d$, which are split by the crystal field of the local Mn–O coordination geometry[32]. A clear loss of intensity on Mn 3$d$-O 2$p$ hybridization feature is observed when the number of Na$^+$ decreases, implying a decrease of Mn oxidation state on the surface, in line with the results from the Mn K-edge XANES spectra (Fig. 3b). The hybridization parameters (defined as absorbance/(e$_g$ holes + 1/4t$_{2g}$ holes)) of Mn–O bonds of Na$_2$Mn$_3$O$_7$ and NaMn$_3$O$_7$ are calculated to be about 0.45 and 0.51[33]. Moreover, the evident shift of the O K pre-edge to a higher energy region further confirms the decrease of Mn oxidation state owing to the increase of oxygen vacancy when the number of Na$^+$ decreases, in agreement with the Mn L-edge XAS spectra (Fig. 3e). As mentioned later, the OH$^-$(aq.) tends to spontaneously fill the oxygen vacancy sites of NaMn$_3$O$_7$ under electrochemical OER conditions. On further electrochemical deprotonation, the lattice oxygen ions coordinated with two Mn ions would begin to be oxidized, producing oxygen hole states in $|O_{2p}$.

**Electrocatalytic measurement for water oxidation**. To demonstrate the benefits of alkali metal-mediation, the as-synthesized materials were evaluated for electrocatalytic water oxidation by the rotating disk electrode technique in O$_2$-saturated 1 M KOH electrolytes (the experimental details are shown in Supplementary Fig. 15 and "Methods"). Figure 4a shows the resistance-($iR$)-corrected polarization curves of Na$_x$Mn$_3$O$_7$, where the currents are normalized by Brunauer–Emmett–Teller (BET) surface areas to reflect the intrinsic activity (Supplementary Fig. 16 and Table 6)[34]. The overpotentials for reaching a specific current density of 0.25 mA cm$^{-2}_{ox}$ are used for activity comparison. Na$_2$Mn$_3$O$_7$ shows the lowest activity with an overpotential of 370 mV. As the value of $x$ decreases, the overpotentials are significantly decreased to 300 and 280 mV for Na$_{1.5}$Mn$_3$O$_7$ and NaMn$_3$O$_7$, respectively. However, further decreasing $x$ significantly lowers the activity, with the high overpotential of 340 mV for Na$_{0.7}$Mn$_3$O$_7$. Figure 4b further compares the specific current densities at a constant overpotential of 320 mV. As expected, NaMn$_3$O$_7$ gives the highest specific current density of 1.08 mA cm$^{-2}_{ox}$, which is 36.0, 2.16, and 12.0 times higher than Na$_2$Mn$_3$O$_7$ (0.03 mA cm$^{-2}_{ox}$), Na$_{1.5}$Mn$_3$O$_7$ (0.45 mA cm$^{-2}_{ox}$) and Na$_{0.7}$Mn$_3$O$_7$ (0.09 mA cm$^{-2}_{ox}$), respectively. Considering the intrinsic activity of the catalyst is controlled by both geometric and electronic properties of active sites, the descriptor of P1–P2 is averaged on different oxygen sites in Na$_x$Mn$_3$O$_7$ (Fig. 4b). As seen, P1–P2 scales with the number of Na$^+$ in a linear manner, which can explain the optimal performance of NaMn$_3$O$_7$, consistent with the predictions from DFT calculations (Fig. 2).

Tafel plots (Fig. 4c) are further derived from the polarization curves of Na$_x$Mn$_3$O$_7$ with the previously benchmarked IrO$_2$ as reference[35]. As seen, the electrocatalytic activity of NaMn$_3$O$_7$ outperforms the benchmarked IrO$_2$ catalyst. Moreover, Na$_x$Mn$_3$O$_7$ (where $x$ = 1.5, 1, and 0.7) shows the decreased Tafel slopes of 54.9, 36.4, and 48.3 mV dec$^{-1}$, respectively, compared with Na$_2$Mn$_3$O$_7$ (61.2 mV dec$^{-1}$), suggesting the smaller

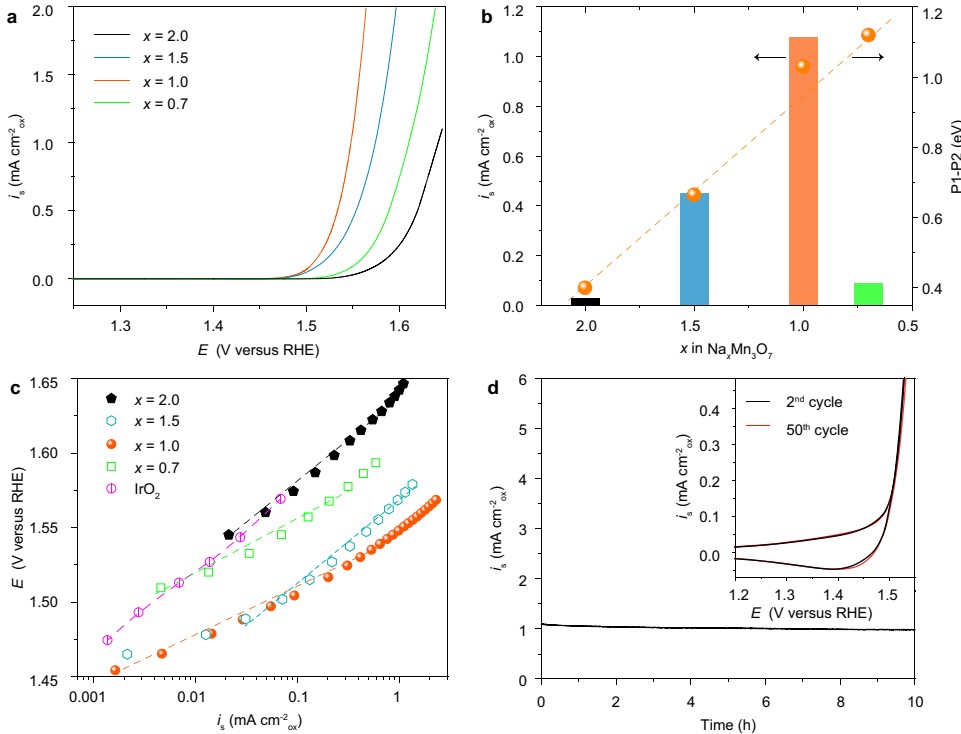

**Fig. 4 Electrocatalytic OER measurements. a** Polarization curves (current normalized by oxide BET surface area) of the as-synthesized $Na_xMn_3O_7$. (The 2st LSV curve is used for comparison.) **b** Comparison of the normalized current densities based on BET surface area at 1.55 V versus RHE (the right y axle shows the relationship between P1 and P2 with the number of $Na^+$ in $Na_xMn_3O_7$.). **c** Tafel plots with the previously benchmarked $IrO_2$[35] as a comparison. **d** Chronoamperometric curve of $NaMn_3O_7$ at 1.55 V versus RHE (The inset shows the 2nd, 10th, 25th, and 50th CV scans of $NaMn_3O_7$).

overpotentials to achieve the same kinetic OER current density. In addition, the Faradaic efficiency of 97% is measured for the best-performing catalyst of $NaMn_3O_7$, indicating that the measured current is primarily originated from water oxidation. The stability measurements were further carried out for $Na_xMn_3O_7$ at a constant overpotential of 320 mV for 10 h. For $NaMn_3O_7$, it maintains 95% of its initial specific activity (Fig. 4d). Similar observations can be made for $Na_{1.5}Mn_3O_7$ and $Na_{0.7}Mn_3O_7$ (Supplementary Fig. 17). Besides, no evident change in pseudocapacitive and OER currents for $NaMn_3O_7$ during cyclic voltammetry (CV) scans (Fig. 4d, inset) were observed as a good indicator of structural stability of the catalysts[28,36]. Neither evident surface amorphization in HRTEM images nor peak variation in XRD patterns was observed for the post-cycled catalyst after OER measurement (Supplementary Fig. 18). The absence of signal from the XPS spectra of K 2p (Supplementary Fig. 19) on the cycled $NaMn_3O_7$ after OER measurement shows that the intercalation of $K^+$ is below the detection limit and negligible. ICP-MS test on the electrolyte further demonstrates that no evident leaching of Na and Mn cations as the OER proceeds on $Na_xMn_3O_7$ ($x$ = 1.5, 1, and 0.7). Actually, Mn ions are unlikely to be able to migrate to the interlayer sites for $NaMn_3O_7$ with the calculated energy barrier as high as 2.80 eV due to the size mismatch between the Mn and Na ions[27,37,38]. Moreover, the charge disproportionation reaction that typically leads to the dissolution of Mn ions is unfavorable in alkaline media[39,40]. As such, we attribute the OER durability and structural stability to the ordered native vacancies in $NaMn_3O_7$ that can self-regulate its deformation and electrochemical reversibility[17,27].

On the contrary, $Na_2Mn_3O_7$ shows evident enhanced activity during OER measurement (Supplementary Fig. 17). ICP-MS test on the electrolyte collected after OER measurement on the $Na_2Mn_3O_7$, demonstrates the evident leaching of $Na^+$ after OER

measurement, whereas the negligible leaching of Mn cations can be found (Supplementary Table 8). The diffraction peak of the post-cycled $Na_2Mn_3O_7$ exhibits a little shift to a higher angle, confirming the interlayer $Na^+$ in the lattice is predominantly leached (Supplementary Fig. 20a). Partial surface amorphization with the thickness of 3–5 nm was also observed from HRTEM image for the post-cycled $Na_2Mn_3O_7$ (Supplementary Fig. 20b). We deduce that the high oxidative OER potential drives the $Na^+$ leaching[38]. As a result, the activated lattice oxygen from the $Na^+$ leaching contributes to the enhanced activity for $Na_2Mn_3O_7$.

**Verification of oxygen evolution mechanism and active site.** Resolving the near-surface structures under electrochemical condition of the catalyst in its highest metastable catalytic state is a prerequisite for the understanding of the OER mechanism and related active site[41]. As such, the in situ X-ray photoelectron spectroscopy (XPS) measurements were performed on $NaMn_3O_7$. From Mn 2p XPS spectra (Fig. 5a), the binding energy shifts to a higher energy at an applied potential of 1.25 V compared with that collected at open circuit, indicating the oxidation state of surface Mn increases. With further increase of potential to 1.55 V, no evident variation of Mn 2p spectra indicates that the Mn ions of the catalyst are structurally and electronically similar to that of 1.25 V. From O 1s XPS spectra, the characteristic peak at 531.2 eV corresponding to oxygen vacancy diminishes at an applied potential of 1.25 V compared with that collected at open circuit. Moreover, the characteristic peak at 529.3 eV corresponding to lattice oxygen shifts to higher energy upon the increase of potential, indicating the oxidation of lattice oxygen. All these indicate that the refilling of oxygen vacancy with $OH^-$ (aq.) and the subsequent deprotonation occur before the electrochemical OER process, in agreement with the DFT calculations and in situ XAS measurement (Fig. 5c and Supplementary Figs. 21, 22). As

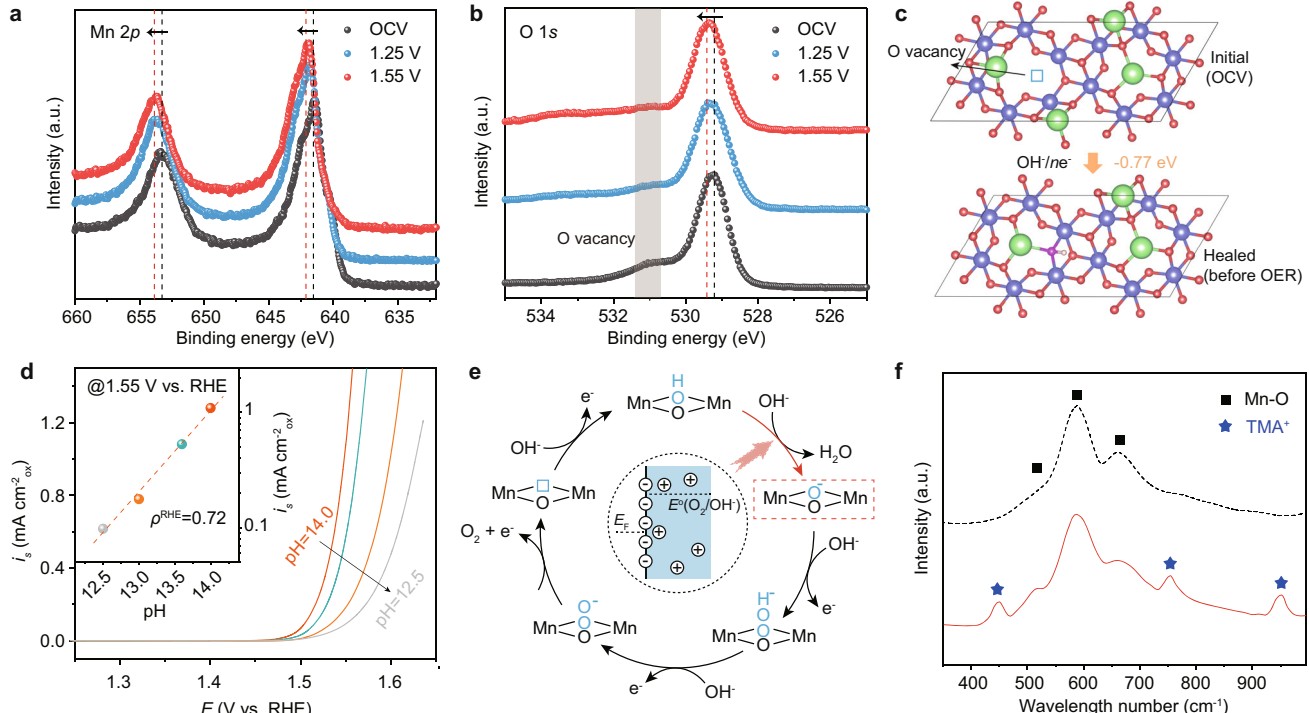

**Fig. 5 Evidence of lattice oxygen as reaction site under OER conditions.** In situ XPS spectra of Mn $2p$ (**a**) and O $1s$ (**b**) of NaMn$_3$O$_7$ under open circuit, 1.25 and 1.55 V (versus RHE). **c** The process of oxygen vacancy refilling under 1.23 V and the associated energy barrier. **d** pH dependence of the OER activities of NaMn$_3$O$_7$. The inset is the proton reaction order estimated by $\rho^{RHE} = (\partial \log(i)/\partial pH)$, with $\rho^{RHE}$ and $i$ being the proton reaction order and current density at a fixed potential of 1.55 V vs. RHE. **e** Schematic OER pathway of acid–base nucleophilic attack involving the rate-limiting proton transfer decoupled from electron transfer step. The inset illustrates the formation of negatively charged oxide surface when equilibrated with the electrolyte. **f** Raman spectra of NaMn$_3$O$_7$ electrodes. These electrodes were respectively operated at 1.55 V versus RHE in 1 M KOH (black line) and TMAOH (red line) solution, then were thoroughly washed by rinsing with high-purity water and acetone for ex situ Raman spectroscopy measurement.

shown in Fig. 5c, OH$^-$ (aq.) tends to spontaneously refill the oxygen vacancy sites of Na$_x$Mn$_3$O$_7$ under 1.23 V vs. RHE. From Pourbaix diagram (Supplementary Fig. 21), the surface termination of Na$_x$Mn$_3$O$_7$ shows the stoichiometric composition under electrochemical OER conditions. Accompanied by the decreased interatomic Mn–O distance from OCV to 1.55 V (Supplementary Fig. 21), the more electrophilic oxygen species with oxygen hole can be formed on the highly covalent oxides after the deprotonation and involve in the subsequent O–O bond formation[8,28]. Combining all these results, we demonstrate that activating lattice oxygen leads to the enhanced OER activity, as the Fermi level enters the $|O_{2p}$ states for Na$_x$Mn$_3$O$_7$ ($x < 2$) due to the charge compensation and redistribution, creating the reactive oxygen radicals on the surface which behave as electrophilic centers prone to nucleophilic attack from the oxygen lone pairs of OH$^-$.

The study of the kinetic isotope effect and pH dependence on OER activity can provide further insight into the reaction mechanism and related intermediates on NaMn$_3$O$_7$[6,42,43]. Because proton mobility in deuterated water solutions can be 1.6–5.0 times slower than that in various protonated water electrolytes, the use of D$_2$O can effectively slow down the proton-transfer kinetics. As shown in Supplementary Fig. 23, the deuterium kinetic isotope effect (at the potential of 1.55 V) is 2.91 for NaMn$_3$O$_7$ in KOH solution with the increased Tafel slope from 36.4 to 65.1 mV dec$^{-1}$, validating the proposed rate-limiting step involves cleavage of the O–H bonds. Such effect is absent in the case of Na$_2$Mn$_3$O$_7$. From pH-dependent OER measurements on RHE scale, NaMn$_3$O$_7$ shows the enhanced activity with the increase of pH from 12.5 to 14. The strong pH dependence indicates the chemical deprotonation step is rate-

limiting (Fig. 5d). Correspondingly, one possible OER pathway on NaMn$_3$O$_7$ is proposed in which the deprotonation of *OH involves only proton transfer (M–OH + OH$^-$ → M–O$^-$ + H$_2$O) and is decoupled from the subsequent electron transfer during the release of oxygen (Fig. 5e). Similar phenomenon is also reported for other highly covalent oxides, in which the oxygen redox generates the negatively charged oxygenated species and therefore results in weak OH$^-$ affinity[42,44]. The partial charge of such species is not directly measurable, but DFT calculations have indeed shown that these intermediates bind to the catalyst surface as a peroxo- or supero-like anion[6,8,42,45]. As seen, the number of the transferred electron from NaMn$_3$O$_7$ to the absorbed intermediate of *OH and *OO is lower than that from Na$_2$Mn$_3$O$_7$, meanwhile the O–O bond distance of 1.299 Å in *OO adsorbed on NaMn$_3$O$_7$ is closer to O$_2^-$ than that of 1.370 Å on Na$_2$Mn$_3$O$_7$ (Supplementary Fig. 24). To track these charged intermediates on NaMn$_3$O$_7$ during OER, tetramethylammonium cation (TMA$^+$) as a chemical probe is introduced to the solution because of its specific electrostatic interaction with negative oxygenated intermediates[8,46]. As expected from the Raman spectra (Fig. 5f), there are three new peaks appear at 451, 753, and 951 cm$^{-1}$, coinciding with the characteristic peaks of TMA$^+$, when the NaMn$_3$O$_7$ electrode was operated at a constant potential of 1.50 V versus RHE in 1 M tetramethylammonium hydroxide (TMAOH) electrolyte. We further compare the OER activities of NaMn$_3$O$_7$ in 1 M KOH and TMAOH solutions (Supplementary Fig. 25). A drop in OER activity with the change of Tafel slope from 48.3 to 50.1 mV dec$^{-1}$ can be observed in the case of TMAOH because of the partial inhibition of the OER, resulted from strong electrostatic interaction between TMA$^+$ and negative oxygenated intermediates.

In summary, through a combination of in-situ spectroscopy-based characterization and first-principles calculations, we have employed modeled $Na_xMn_3O_7$ materials to illustrate the crucial roles of alkali metal mediation to tune lattice oxygen reactivity and scaling relation for the rational design of better OER electrocatalysts. More specifically, an intermediate level of $Na^+$ mediation ($NaMn_3O_7$) manifests the optimal activity, which is due to the regulation of the barrier symmetry between O–H bond cleavage and *OOH formation on the basis of overcoming the preexisting scaling relation. In addition, the pH-dependent experiment and Raman spectra further demonstrate it works in a decoupled proton/electron route with the presence of negatively charged oxidized oxygen species. This work provides a guideline for the rational design of better catalysts towards electrocatalytic water oxidation or other oxidative reactions through tuning lattice oxygen reactivity and scaling relation.

## Methods

**Synthesis of $Na_xMn_3O_7$.** $Na_xMn_3O_7$ ($x = 2.0$, 1.5, 1.0 and 0.7) materials were synthesized by a modified method of solid-state reaction[17,27,47]. In detail, the starting materials of $NaNO_3$ and $MnCO_3$ with the desired Na/Mn ratios were thoroughly mixed in an agate mortar and pressed into pellets under pressure of 10 MPa. Then, the $Na_xMn_3O_7$ materials can be obtained by calcining such pellets at 500–650 °C in tube furnace with $O_2$ atmosphere for 5–10 h. Before use, the $Na_xMn_3O_7$ materials were stored in a glovebox with Ar atmosphere.

**Composition and structure characterization.** X-ray diffraction (XRD) patterns were collected with a Bruker D8 FOCUS equipped with nickel-filtered Cu Kα radiation ($\lambda = 1.541$ Å). Field emission SEM characterization was performed with a Hitachi S-4800 SEM. TEM and EDS characterizations were performed with a JEM-2100F transmission electron microscope. Elemental composition was analyzed using the techniques of Vista-MPX EL02115765 Inductively coupled plasma spectrometry and PerkinElmer NexION 350× Inductively coupled plasma-mass spectrometry (ICP-MS). For ICP-MS measurement, the standard curve is linearly fitted in the range of 0.1, 1.0, 10.0, 100, 500, and 1000 ppb with the internal standard of Rh ($10\,\mu g\,L^{-1}$). The BET surface areas were obtained from $N_2$ sorption isotherm measurements on Micrometrics TriStar 3000 equipment.

Mn K-edge X-ray absorption spectra (XAS) were collected at the X-ray Absorption Fine structure for catalysis (XAFCA) beamline at the Singapore Synchrotron Light Source (SSLS) using the transmission mode[48]. The photon energy is ranged from 1.2 to 12.8 keV achieved by two sets of monochromator crystals of Si (111) and $KTiOPO_4$ crystal. In-situ XAS measurements were conducted with a home-made electrochemical cell in fluorescence mode. The fluorescence yields were collected with silicon drift detector (Bruker Xflash 6|100). The catalysts were coated on carbon paper with mass loading of $2\,mg\,cm^{-2}$ as the working electrode. Acquired EXAFS data were analyzed using ATHENA module implemented in the IFEFFIT software packages[49]. The k- and R-ranges to fit the EXAFS data were set as 2–12 $Å^{-1}$ and 1.0–3.5 Å, respectively. Mn L-edge and O K-edge XAS were collected at Soft X-ray-ultraviolet (SUV) beamline at SSLS. X-ray photoelectron spectroscopy (XPS) characterizations were performed on a Thermo ESCALAB 250Xi X-ray photoelectron spectroscope using the home-made X-ray cell. At different applied potentials, the working electrodes were first stabilized to reach a steady state, then the corresponding XPS signals were collected and analyzed.

**Electrochemical measurements.** Electrochemical measurements were conducted in a three-electrode setup with graphite rod and Hg/HgO (1 M KOH) as the counter and reference electrode. The working electrode was prepared by coating catalyst ink on a glassy carbon with mass loading of $0.204\,mg_{ox}\,cm^{-2}$. Typically, 4 mg catalyst and 0.8 mg acetylene black were suspended in 2 mL mixture solution of isopropanol, water and Nafion and violently ultrasounded for 3 h to form a homogeneous ink. To decrease the influence of capacitive current and gas bubbles, the linear sweep voltammetry measurements were performed with the scan rate of $2\,mV\,s^{-1}$ and rotational speed of 1600 r.p.m. The Tafel plots were derived from the polarization curves as the function between overpotential and the log current ($\eta = b\log[J] + a$), where $b$ represents as Tafel slope. CV measurements were conducted at a scan rate of $10\,mV\,s^{-1}$ to investigate the pseudocapacitive charge preceding the OER region. All the used potentials were calibrated based on the RHE. The Faradic efficiency was measured using in-line gas chromatograph, which is defined as the ratio between the amount of experimentally measured $O_2$ and the amount of theoretically produced $O_2$ from the reaction.

**Computational details.** Spin-polarized DFT calculations were performed on Vienna ab initio Simulation package with projector augmented wave pseudopotential and revised Perdew–Burke–Ernzerh functional[50–53]. The kinetic energy cut-off was set to 520 eV. To better describe the localized $3d$ orbital, the effective U value of Mn was set to 3.9 eV[38]. For structure optimization, the Brillouin zone was sampled by Gamma-centered k-point with $5 \times 5 \times 1$ ($3 \times 3 \times 1$) in $2 \times 2$ ($3 \times 3$) supercell. For electronic structure calculation, the $3 \times 3 \times 1$ ($3 \times 1 \times 1$) k-point was used in $1 \times 1$ ($1 \times 2$) supercell. The size of such supercell is suitable for accurate calculation of surface OER reaction (Supplementary Table 9). The convergence criterion of force and energy were set to 0.02 eV $Å^{-1}$ and $10^{-5}$ eV, respectively. For vdW correction, DFT-D3 method with Becke–Jonson damping was used[54,55]. For search of transition state, the climbing image nudged elastic band (CI-NEB) method was used[56,57].

## Data availability
The data that support the findings of this study are available from the corresponding author upon reasonable request. Source data are provided with this paper.

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

## Acknowledgements

The authors appreciate the support from the National Key R&D Program of China (2020YFA0710000), National Natural Science Foundation of China (22008170), and National Research Foundation (NRF), Prime Minister's Office, Singapore, under its Campus for Research Excellence and Technological Enterprise (CREATE) program. We also acknowledge financial support from the academic research fund AcRF tier 1 (M4012076 RG118/18), Ministry of Education, Singapore, AME Individual Research Grant (Grant number: A1983c0026), Agency for Science, Technology, and Research (A*STAR), Singapore. Additionally, the authors appreciate the XAS measurements from SSLS, SUV (Soft X-Ray-Ultraviolet) beamline, and XAFCA beamline.

## Author contributions

X.W. and Z.-F.H. proposed the studies and wrote the paper. Z.-F.H. synthesized the materials and performed the electrocatalytic tests. Z.-F.H., J.S., and Z.J.X. performed the density functional theory calculations. Z.-F.H., S.D., and X.L. conducted XRD, SEM, TEM, XPS, and other characterizations. S.X., Y.D., and C.D. conducted XAFS measurements. All authors discussed the results and commented on the manuscript.

## Competing interests

The authors declare no competing interests.
