## [Peer Review File · Nature Communications]

REVIEWER COMMENTS

Reviewer #1 (Remarks to the Author):

This manuscript employs both theoretical and experimental methods to show that an intermediate level of Na⁺ ions in Na_xMn₃O₇ leads to the optimum OER activity. The authors claim that the number of Na ions in the catalyst can be related to the number of Mn vacancies, which in turn affects the lattice oxygen reactivity. They also claim that the presence of Na on the catalyst surface particularly stabilizes OOH* via non-covalent bonding, breaking linear scaling relations to result in NaMn₃O₇ being more active than the conventionally predicted optimum catalyst, IrO₂.

I believe that the work is of potential interest to the field, and can be considered as publishable in the journal, but only after the authors adequately address the following major concerns.

(1) The nominal charge of the Na_xMn₃O₇ (x=1.5, 1.0, 0.5) surface considered in this study is negative. Thus, other (charge-neutral) surfaces such as those containing lattice/surface oxygen vacancies can be more relevant under the OER condition. The authors need to provide a pourbaix diagram, and show that, even with the extra negative charge, the Na_xMn₃O₇ (x=1.5, 1.0, 0.5) surface without any oxygen vacancies is indeed the relevant surface.

(2) In Table S4, we can see that G1 and G2 values, which are not related to OOH* stability at all, vary significantly among S1 ~ S6 models. This can not be explained by the noncovalent interaction between Na and OOH.

(3) The authors need to show separately how O*, OH*, and OOH* adsorption energies are affected by the number of Na ions, and confirm their argument that OOH* is indeed selectively stabilized by the Na ions. In addition, they need to show clearly how OOH* vs OH* scaling relation is broken by their systems.

(4) The color codes used in many figures are either omitted or not very distinguishable. Generally, more detailed descriptions of figures are required. Typos and grammar mistakes can be noticed frequently.

Reviewer #2 (Remarks to the Author):

Current work is a fundamental study on the alkaline oxygen evolution reaction (OER) electrocatalysis using layered oxides. The authors suggest that the electrocatalytic activity of layered sodium manganese oxides can be controlled by the amount of Na⁺ cations in the oxide. Based on a very thorough theoretical calculations and X-ray spectroscopy analysis, it is suggested that the lattice oxygen reactivity (related to electronic and geometric parameters) is responsible for the overall OER activity. The work is concise and very well written. The selection of the theoretical and experimental procedures is optimal, although some optimization is possible, e.g. ICP-MS instead of ICP-OES – see below. In general, the authors conclusions are supported by the results. Hence, I recommend publication of this manuscript after a major revision in the light of the comments presented below. My comments are mostly directed to the section describing the electrochemical results. To evaluate DFT and XAS data I would suggest an extra round of review.

Comments:

1. My main concern is to the part describing stability of these materials. Layered sodium manganese oxides are well-known as cathode materials in Na-ion batteries. This implies that Na⁺ ions can intercalate/de-intercalate depending on the potential applied to the electrode. Moreover, according to

recent studies, also K⁺ ions can enter/leave the lattice of such layered material - <https://pubs.acs.org/doi/10.1021/acsaem.8b01016>. The latter is of relevance, as K⁺ based electrolyte is used in the current work. Indeed, potassium cation intercalation was suggested very recently in another work published in this journal - <https://www.nature.com/articles/s41467-020-15231-x>. Hence, I (and I also believe readers) would like to see the authors comments and discussion addressing these topics.

2. Continuing with stability, I am puzzled with the ICP-OES data. The authors write that after performing a 10 hours stability test, no Na or Mn is found in the electrolyte. On the other hand, after XAS analysis, 20% of Na is dissolved. The authors should discuss why dissolution of Na is so different in two cases.

3. ICP-OES is some 3-4 orders of magnitude less sensitive than ICP-MS. In case ICP-OES does not show Na/Mn, it does not mean that these materials are not dissolving, it means that ICP-OES cannot see them. Hence, I would recommend to use ICP-MS.

4. Even for ICP-OES after XAS, some Mn is found in the electrolyte. Although, the amount is low. Recently it was shown that Mn can dissolve during the OER - <https://pubs.acs.org/doi/abs/10.1021/acs.jpcc.9b07751>. The authors should comment on the high stability of their material. Can it be that the loss of Na⁺ and intercalation/deintercalation of K⁺ as suggested here <https://www.nature.com/articles/s41467-020-15231-x> stabilizes Mn against dissolution?

5. I find it good that the authors quantify activity showing numbers. Unfortunately, stability is not quantified. For instance, the authors write "... the Faradaic efficiency of near-unity is measured for the best-performing catalyst of NaMn3O7, indicating that the measured current is primarily originated from the water oxidation.". This is obviously not enough. 0.9 is also near unity but means a massive loss of material assuming that the rest 10% is due to dissolution. Hence, the exact value must be given and the reason for not having FE=100% must be discussed.

6. The authors write "negligible loss of specific activity" – how much exactly? In the way how it shown in Figure 4d, it is difficult to see but it is still visible that the activity drops.

7. Stability data should be shown for all materials, not just for one.

8. Probably a specialist can just this better but for me it seems that Figure S16 does show some amorphization as one would expect based on the ICP-OES data after the XAS experiments (deintercalation without amorphization should be also possible, needs to be discussed).

9. According to authors, 20% of Na is lost during the XAS test. Assuming that Na dissolves completely from the surface layer, calculate the thickness of the Na-free layer. Why TEM does not show this?

10. Assuming that there is a gradient in Na⁺ concentration with thickness (0 at surface and nominal value in the bulk), how does this change the conclusions from XAS data and mechanistic analysis?

11. Dissolution data is shown for only one material. Data for other samples should be shown.

12. The authors write "Recent discoveries have demonstrated that the surface metal electrochemical leaching is responsible for partial lattice oxygen loss ($\text{Na}_2\text{Mn}_3\text{O}_7 + (1-x/2)\text{H}_2\text{O} \rightarrow (2-x)\text{Na}^+ + \text{Na}_x\text{Mn}_3\text{O}_{6+x/2} + (2-x)\text{OH}^-$ ". According to this equation, Na⁺ dissolves. This must be stated but also discussed.

13. Is electrochemical surface area changing with Na⁺ leaching? If so, how much and how it influences activity?

14. The authors write that "the post-cycled Na₂Mn₃O₇ shows the enhanced OER activity than the fresh one (Supplementary Fig. 18d), due to the activation of lattice oxygen from the Na⁺ leaching." This needs to be elaborated more. The main idea of this work that there is an optimal amount of Na. Na=1 is shown to be the most stable. Assuming that Na is leaching from the Na=2 sample, it makes sense that activity is improving. On the other hand, with Na<1, activity is decreasing. Does it mean that eventually this material will lose its activity with more and more Na dissolved?

Reviewer #3 (Remarks to the Author):

Huang et al. report Na_xMn₃O₇ as a model system to study the mechanism of the oxygen evolution

reaction (OER), particularly, the role of oxygen reactivity. The catalysts are studied by DFT, XRD, multi-edge XAS, HRTEM, TEM-EDS, ICP-OES, gas adsorption, gas chromatography, Raman spectroscopy and electrochemical methods.

The best catalyst in the series is very active for a manganese oxide and is reasonably stable for 10 h at low overpotential. Mn is also non-toxic and abundant (as is Na). This makes the mechanistic study noteworthy and interesting for the field of electrocatalysis, in particular green hydrogen production. The main claim is that the number of Na⁺ correlates with the number of oxygen holes, which seems to be supported by the theoretical calculations (not my area of expertise) but I have doubts about the experimental proofs of the claim (which falls within my area of expertise) as detailed below. Additional evidence and analysis is needed to support the claim(s) as also detailed below. The manuscript is difficult to follow as observations are not clearly separated from interpretations and interpretations are not or too briefly discussed. There are technical issues, particularly with the XAS and electrochemical evaluations. On balance, the topic of lattice oxygen reactivity and the materials system are interesting and the role of alkali ions clearly deserves more attention. However, many additional experiments and discussion are needed to strengthen the manuscript for publication.

Please address:

1) Claims

a) The presented data do not clearly support the involvement of lattice oxygen. The in situ XAS do not support that the Mn valence remains unchanged. Are changes expected in the in situ XAS for rods of 200 nm width and 800 nm length (Suppl. Fig. 13)? How much does the surface contribute to the signal? I doubt that Mn-K edge XAS is suited to resolve surface changes, which is also supported by the clearer changes in soft X-ray XAS in Fig. 3. A more surface-sensitive method, e.g. soft XAs or XPS is required.

b) I am wondering how controlled the active state is when Na leaches from the as-synthesized material. Do all leach Na and take up H⁺? The elucidation of the composition of all catalysts is needed to support that the active state can be rationally designed.

2) Please separate observations from interpretation and provide additional information so that the interpretations can be followed by non-experts

a) What is the significance of the magnetization moments of the oxygen ions? Why is it important for catalysis? What is the relation to the parameter P1?

b) How are the parameters P1 and P2 calculated? Why has ideal catalysis the values in Suppl. Table 2? I would have assumed they are zero?

c) Why should Na leaching activate lattice oxygen? It needs more discussion and is an interpretation.

d) What is the conclusion of the Raman experiments? It needs discussion. Does it indicate that TMAOH intercalates?

3) XAS analysis

a) Fig. 3c does not support the interpretation of its FT in Fig. 3d. It appears that x=2 and x=1 have very similar wavenumbers where y=0 (meaning similar distance) and similar amplitude for low wavenumber (meaning similar coordination number of Mn-O). The discussion of interatomic distances and coordination numbers requires EXAFS fits., preferably in k-space.

b) The x-axis labels in Fig. 3d and Fig. 5b are misspelled. It is not a "radical" but a "radial" distance. Actually, reduced distance would be most appropriate as there is not mentioning that the authors corrected the axis of the phase shifts.

c) why are only x=2 and x=1 shown in Fig. 3d,e,f,? What about the other samples?

d) Fig. 3e does not show the O-K main edge. No conclusions can be made about it. The pre-edge is shown. Please discuss why the shift is fully assigned to the oxygen valence and not to changes of the Mn ligand field. Are there other reports in literature where oxygen holes are studied by the OK pre-edge?

e) Three reference materials are insufficient for a good calibration curve for the Mn valence. Furthermore, the edge position depends on the type of ligand. Mn as the ligand of the 1st coordination shell is drastically different as compared to O.

f) The Mn-O bond distance of 1.2 Å is much too short. This is the reduced distance. Even when the

phase shifts are accounted for, the distance is still too short. Please compare it to the Mn-O distances from the $\text{Na}_x\text{Mn}_3\text{O}_7$ crystal structure. Perhaps an EXAFS fit gives a more realistic value.

g) Please quantify the "hybridization feature" in Fig. 3e. Can the change be explained by Mn oxidation?

4) Analysis of the Mn and O valences

a) Can it be excluded that proton intercalation occurs rather than oxygen oxidation? If so how?

b) The formulation "This counterintuitive phenomenon could be rationalized by the possible lattice oxygen losses due to the increased Mn-O covalency" Is misleading as it is later discussed that both the number of O and their valence may change.

5) Introduction. Catalytic mechanisms based on lattice oxygen should be mentioned explicitly in the introduction. Both nucleophilic attack and O-O coupling are possible without the involvement of lattice oxygen

6) Lattice oxygen must be defined. Suppl. Fig. 1 shows (a) μ^3 -oxo, (b) μ^2 -oxo, (c) μ^1 -oxo motifs. Only (a) and (b) are lattice oxygen.

7) DFT calculations

a) Which motif was the active site in the DFT calculations, μ^3 -oxi (Suppl. Fig. 1a) or μ^2 -oxo (Suppl. Fig. 1b)?

b) How would it change the conclusions of the DFT results if Na^+ was partially replaced by H^+ as suggested by the experimental results?

c) Are the clusters sufficiently large to exclude size effects, e.g. due to mirror charges?

8) descriptor P1-P2

a) how transferable is the descriptor P1-P2 to other materials classes/families?

b) do both P1 and P2 scale linearly with Na?

c) why is P1 positive and P2 negative?

9) why is the reaction order fractional? Was it ensured that the intercalation of H^+ ceased before the catalytic investigation?

10) Minor

a) why evaluation at 0.25 mA/cm²_ox. Is it also used elsewhere?

b) Fig. 5c is not clear. It looks like a terminal O is healed on the right but it is a μ^2 -oxo bridge on the left.

c) Fig. 5e is not clear. On which oxygen sites does OH adsorb? A μ^2 site? If so, no $\sim 180^\circ$ M-O-M angle should be drawn.

We thank the reviewers for their time and very useful comments in improving the quality of this manuscript. Provided below is our detailed point-to-point response to each question. The changes in the manuscript have been highlighted and listed below.

Reviewer #1

This manuscript employs both theoretical and experimental methods to show that an intermediate level of Na^+ ions in $\text{Na}_x\text{Mn}_3\text{O}_7$ leads to the optimum OER activity. The authors claim that the number of Na ions in the catalyst can be related to the number of Mn vacancies, which in turn affects the lattice oxygen reactivity. They also claim that the presence of Na^+ on the catalyst surface particularly stabilizes OOH^* via non-covalent bonding, breaking linear scaling relations to result in NaMn_3O_7 being more active than the conventionally predicted optimum catalyst, IrO_2 . I believe that the work is of potential interest to the field, and can be considered as publishable in the journal, but only after the authors adequately address the following major concerns.

Comment 1:

The nominal charge of the $\text{Na}_x\text{Mn}_3\text{O}_7$ ($x=1.5, 1.0, 0.5$) surface considered in this study is negative. Thus, other (charge-neutral) surfaces such as those containing lattice/surface oxygen vacancies can be more relevant under the OER condition. The authors need to provide a pourbaix diagram, and show that, even with the extra negative charge, the $\text{Na}_x\text{Mn}_3\text{O}_7$ ($x=1.5, 1.0, 0.5$) surface without any oxygen vacancies is indeed the relevant surface.

Response 1:

Thanks for the reviewer's comment. In DFT calculations, we do use the charge-neutral surfaces of $\text{Na}_x\text{Mn}_3\text{O}_7$ (stoichiometric composition of $\text{Na}_{2x/7}(\text{Mn}_{6/7}\square_{1/7})\text{O}_2$, \square represents Mn vacancy, $x = 2, 1.5, 1$ and 0.5) slabs (see Supplementary Fig. 3), not the charge-negative surfaces. As known, Mn would be in their maximum achievable oxidation states of 4+ in all these modeled surfaces. When the number of Na^+ is decreased from 2 to 0.5, the charge neutrality is tuned by lattice oxygen oxidation with the generation of oxygen holes. Such oxygen holes can be confirmed from the projected density of states and partial charge density near E_F (Fig. 1b, c and Supplementary Table 1).

It is true that lattice oxygen oxidation typically produces oxygen vacancies in metal oxides, which is confirmed from Mn K-edge XANES spectra (Fig. 3b, d). The *in-situ* XAS measurements (Fig. 5a, b) under electrochemical OER conditions further confirm the fast refilling of oxygen vacancy to recover the charge-neutral surfaces of $\text{Na}_x\text{Mn}_3\text{O}_7$ with stoichiometric composition. As shown in Fig. 5c, OH^- (aq.) actually tends to spontaneously refill the oxygen vacancy sites of $\text{Na}_x\text{Mn}_3\text{O}_7$ under 1.23 V vs. RHE (-0.77 eV). We also provide a pourbaix diagram to demonstrate the stability of surface termination under different potentials (Supplementary Fig. 23). As seen, the positive formation energy of oxygen vacancy can be observed for $\text{Na}_x\text{Mn}_3\text{O}_7$ ($x=1.5, 1.0$ and 0.5) when the potential is higher 1.23 V, indicating the unfavorable formation of oxygen vacancy. Therefore, we believe, the modelled charge-neutral surfaces of $\text{Na}_x\text{Mn}_3\text{O}_7$ with

stoichiometric composition are indeed the relevant surfaces under electrochemical OER conditions that are reasonable to predict the relative activity in experimental.

Change 1:

1). Page 12, line 10, one sentence is added as follows.

This is further confirmed by Pourbaix diagram (Supplementary Fig. 23), which revealed the surface termination of $\text{Na}_x\text{Mn}_3\text{O}_7$ with stoichiometric composition under electrochemical OER conditions.

2). Supporting information, one figure is added as follows.

Supplementary Fig. 23 | Formation energy of oxygen vacancy in $\text{Na}_x\text{Mn}_3\text{O}_7$ ($x = 1.5, 1$ and 0.5).

Comment 2:

In Table S4, we can see that G1 and G2 values, which are not related to OOH* stability at all, vary significantly among S1 ~ S6 models. This cannot be explained by the noncovalent interaction between Na and OOH.

Response and change 2:

We thank the reviewer for the critical comment. We are sorry that Table S3 and Table S4 were reversed in Supplementary Information and we have corrected this mistake in the revised version. Our calculation indeed shows that G1 and G2 values, which are not related to OOH* stability at all, vary significantly among S1 ~ S6 models. The stabilization of OOH* on modelled $\text{Na}_x\text{Mn}_3\text{O}_7$ can be achieved by both electronic effect (i.e., the influence from the oxygen holes) and geometric effect (i.e., the noncovalent interaction between Na and OOH). These two effects are directly determined by the number of Na^+ . We define two parameters (P1 and P2) to separate the mentioned two effects to stabilize *OOH ($\Delta G'^{*}\text{OOH} = \Delta G^*\text{OOH} + \text{P1} + \text{P2}$). To correlate the influence from oxygen hole, the electronic parameter (P1) is defined as the variation of $\Delta G^*\text{O} - \Delta G^*\text{OH}$ with reference to S1. To demonstrate Na^+ -specific noncovalent interaction with *OOH, the geometric parameter (P2) is then defined as the variation of $\Delta G^*\text{OOH}$ with reference to S6. In Table S3, we fix $\text{P2} = 0$ for S1-S6, aiming to demonstrate the role of electronic effect on overpotential. In

Table S4, we fix P1=0 for S1-S6, aiming to demonstrate the role of geometric effect on overpotential (η) and overpotential ceiling (η_{\min}).

Comment 3:

The authors need to show separately how O*, OH*, and OOH* adsorption energies are affected by the number of Na ions, and confirm their argument that OOH* is indeed selectively stabilized by the Na ions. In addition, they need to show clearly how OOH* vs OH* scaling relation is broken by their systems.

Response 3:

We have provided a table (Supplementary Table 5) to show separately how O*, OH*, and OOH* adsorption energies are affected by the number of Na ions. As seen, ΔG^*_{OOH} has no clear correlation with the number of Na ions. This is because the number of Na ions regulates both electronic effect (P1) and geometric effect (P2) to determine the overall adsorption energy of oxygen-containing intermediates ($\Delta G^*_{\text{OH}} = \Delta G^*_{\text{OH}} + P1$; $\Delta G^*_{\text{O}} = \Delta G^*_{\text{O}} + 2P1$; $\Delta G^*_{\text{OOH}} = \Delta G^*_{\text{OOH}} + P1+P2$). Therefore, we cannot conclude that OOH* is selectively stabilized by the Na ions. Actually, we conclude that the specific noncovalent interaction between alkali metal and the pendant oxygen in *OOH is selective to interact with *OOH, not *O and *OH. As a result, the adsorption-energy scaling relation between *OOH and *OH can be broken or regulated according to $\Delta G^*_{\text{OOH}} = \Delta G^*_{\text{OH}} + 3.24+P2$ (left branch of volcano plot shown in Fig. 2d). As seen, the greater number of Na⁺ around lattice oxygen, the lower value of P2, leading to the decrease of overpotential ceiling from 0.39 to 0.19.

Change 3:

1). Supporting information, one table is added as follows.

Supplementary Table 5 | Data for free energy of ΔG^*_{OH} , ΔG^*_{O} , ΔG^*_{OOH} in various models for Na_xMn₃O₇.

Model	ΔG^*_{OH} (eV)	ΔG^*_{O} (eV)	ΔG^*_{OOH} (eV)
S1	0.61	1.28	3.45
S2	0.51	1.75	3.35
S3	0.34	1.78	3.46
S4	0.46	2.02	3.56
S5	0.71	2.35	3.83
S6	0.87	2.68	4.11

Comment 4:

The color codes used in many figures are either omitted or not very distinguishable. Generally, more detailed descriptions of figures are required. Typos and grammar mistakes can be noticed frequently.

Response and Change 4:

We thank the reviewer for the valuable suggestions. To make the manuscript clearer for the readers, we have revised the related figures and checked the typos and grammar mistakes carefully.

Reviewer #2

Current work is a fundamental study on the alkaline oxygen evolution reaction (OER) electrocatalysis using layered oxides. The authors suggest that the electrocatalytic activity of layered sodium manganese oxides can be controlled by the amount of Na⁺ cations in the oxide. Based on a very thorough theoretical calculations and X-ray spectroscopy analysis, it is suggested that the lattice oxygen reactivity (related to electronic and geometric parameters) is responsible for the overall OER activity. The work is concise and very well written. The selection of the theoretical and experimental procedures is optimal, although some optimization is possible, e.g. ICP-MS instead of ICP-OES – see below. In general, the authors conclusions are supported by the results. Hence, I recommend publication of this manuscript after a major revision in the light of the comments presented below. My comments are mostly directed to the section describing the electrochemical results. To evaluate DFT and XAS data I would suggest an extra round of review.

Comment 1:

My main concern is to the part describing stability of these materials. Layered sodium manganese oxides are well-known as cathode materials in Na-ion batteries. This implies that Na⁺ ions can intercalate/de-intercalate depending on the potential applied to the electrode. Moreover, according to recent studies, also K⁺ ions can enter/leave the lattice of such layered material -<https://pubs.acs.org/doi/10.1021/acsaem.8b01016>. The latter is of relevance, as K⁺ based electrolyte is used in the current work. Indeed, potassium cation intercalation was suggested very recently in another work published in this journal - <https://www.nature.com/articles/s41467-020-15231-x>. Hence, I (and I also believe readers) would like to see the authors comments and discussion addressing these topics.

Response 1:

We thank the reviewer for the comment. It is true that the alkali metal ions including Li⁺, Na⁺ and K⁺ ions can enter/leave (intercalate/de-intercalate) the lattice of some layered materials as reported in recent studies (*ACS Appl. Energy Mater.* **2018**, 1, 10, 5410; *Nat. Commun.* **2020**, 11, 1378). In our study, we perform the XPS measurement of K 2p on Na_xMnO₃ (taking Na₂MnO₃ and NaMn₃O₇ as examples) after OER measurement (deionized water washing for three times) and the absence of signal indicates the intercalation of K⁺ is below the detection limit and negligible. We deduce that the size mismatch between Na⁺ (102 pm) and K⁺ (138 pm) leads to that K⁺ cannot intercalate into the lattice of Na_xMnO₃ to occupy the pristine Na⁺ site.

Change 1:

1). **Page 11, line 2**, one sentence is added as follows.

The absence of signal from the XPS spectra of K 2p (**Supplementary Fig. 18**) on the cycled NaMn₃O₇ after OER measurement shows that the intercalation of K⁺ is below the detection limit and negligible.

2). **Supporting information**, one figure is added as follows.

Supplementary Fig. 18 | XPS spectra of K 2p of $\text{Na}_2\text{Mn}_3\text{O}_7$ and NaMn_3O_7 after OER measurement.

Comment 2:

Continuing with stability, I am puzzled with the ICP-OES data. The authors write that after performing a 10 hours stability test, no Na or Mn is found in the electrolyte. On the other hand, after XAS analysis, 20% of Na is dissolved. The authors should discuss why dissolution of Na is so different in two cases.

Response 2:

We thank the reviewer for the comment. For $\text{Na}_2\text{Mn}_3\text{O}_7$, some Na^+ can be dissolved into the electrolyte and the molar ratio of Na/Mn decreases. For the best NaMn_3O_7 material, no Na is further dissolved from the XAS analysis. To fairly compare the leaching, in the revised version, we have provided the stability and dissolution data for all materials at 1.55 V for 10 h. ICP-MS (Supplementary Table 8) on the electrolyte further demonstrates that no evident leaching of Na and Mn cations as the OER proceeds on $\text{Na}_x\text{Mn}_3\text{O}_7$ ($x = 1.5, 1$ and 0.7). For $\text{Na}_2\text{Mn}_3\text{O}_7$, the evident leaching of Na^+ can be observed after OER measurement, whereas the negligible leaching of Mn cations can be observed. We think the removal of Na^+ ions for $\text{Na}_2\text{Mn}_3\text{O}_7$ depends on the potential applied to the electrode. For $\text{Na}_x\text{Mn}_3\text{O}_7$ ($x = 1.5, 1$ and 0.7), the required potentials for removal of Na^+ ions are higher than the OER potential.

Change 2:

1). Page 10, line 14, two paragraphs are revised as follows.

The stability measurements were further carried out for $\text{Na}_x\text{Mn}_3\text{O}_7$ at a constant overpotential of 320 mV for 10 h. For NaMn_3O_7 , it maintains 95% of its initial specific activity (Fig. 4d). Similar observations can be made for $\text{Na}_{1.5}\text{Mn}_3\text{O}_7$ and $\text{Na}_{0.7}\text{Mn}_3\text{O}_7$ (Supplementary Fig. 16). Besides, no evident change in pseudocapacitive and OER currents for NaMn_3O_7 during CV scans (Fig. 4d, inset) were observed as a good indicator of structural stability of the catalysts^{28,37}. Neither evident surface amorphization in HRTEM images nor peak variation in XRD patterns was observed for the post-cycled catalyst after OER measurement (Supplementary Fig. 17). The absence of signal from the XPS spectra of K 2p (Supplementary Fig. 18) on the cycled NaMn_3O_7 after OER measurement shows that the intercalation of K^+ is below the detection limit and negligible. ICP-MS on the electrolyte further demonstrates that no evident leaching of Na and Mn

cations as the OER proceeds on $\text{Na}_x\text{Mn}_3\text{O}_7$ ($x = 1.5, 1$ and 0.7). Actually, Mn ions are unlikely to be able to migrate to the interlayer sites for NaMn_3O_7 with the calculated energy barrier as high as 2.80 eV due to the size mismatch between the Mn and Na ions^{27,38,39}. Moreover, the charge disproportionation reaction that typically leads to the dissolution of Mn ions is unfavorable in alkaline media^{40,41}. As such, we attribute the OER durability and structural stability to the ordered native vacancies in NaMn_3O_7 that can self-regulate its deformation and electrochemical reversibility^{17,27}.

On the contrary, $\text{Na}_2\text{Mn}_3\text{O}_7$ shows the evident enhanced activity during OER measurement (Supplementary Fig. 16). ICP-MS test on the electrolyte collected after OER measurement on the $\text{Na}_2\text{Mn}_3\text{O}_7$, demonstrates the evident leaching of Na^+ after OER measurement, whereas the negligible leaching of Mn cations can be found (Supplementary Table 8). The diffraction peak of the post-cycled $\text{Na}_2\text{Mn}_3\text{O}_7$ exhibits a little shift to a higher angle, confirming the interlayer Na^+ in the lattice is predominantly leached (Supplementary Fig. 19a). Partial surface amorphization with the thickness of 3-5 nm was also observed from HRTEM image for the post-cycled $\text{Na}_2\text{Mn}_3\text{O}_7$ (Supplementary Fig. 19b). We deduce that the oxidative OER potential drives the Na^+ leaching³⁹. As a result, the activated lattice oxygen from the Na^+ leaching contributes to the enhanced activity for $\text{Na}_2\text{Mn}_3\text{O}_7$.

Comment 3:

ICP-OES is some 3-4 orders of magnitude less sensitive than ICP-MS. In case ICP-OES does not show Na/Mn, it does not mean that these materials are not dissolving, it means that ICP-OES cannot see them. Hence, I would recommend to use ICP-MS.

Response and Change 3:

Thanks for the valuable suggestion from the reviewer. We have repeated the related measurement using ICP-MS considering the higher sensitivity than ICP-AES. It is found that the measured Na/Mn mole ratio in $\text{Na}_x\text{Mn}_3\text{O}_{7-\delta}$ using these two different techniques are quite close. The results are shown in revised Supplementary Table 6.

Supplementary Table 6 | Na/Mn mole ratio (y , measured from ^aICP-OES and ^bICP-MS) and BET surface area (SA) of $\text{Na}_x\text{Mn}_3\text{O}_{7-\delta}$.

x in $\text{Na}_x\text{Mn}_3\text{O}_{7-\delta}$	^a y	^b y	SA (m^2g^{-1})
2.0	0.66	0.66	1.10
1.5	0.50	0.51	1.42
1.0	0.34	0.33	1.55
0.70	0.23	0.24	1.98

Comment 4:

Even for ICP-OES after XAS, some Mn is found in the electrolyte. Although, the amount is low. Recently it was shown that Mn can dissolve during the OER -<https://pubs.acs.org/doi/abs/10.1021/acs.jpcc.9b07751>. The authors should comment on the high stability of their material. Can it be that the loss of Na⁺ and intercalation/deintercalation of K⁺ as suggested here <https://www.nature.com/articles/s41467-020-15231-x> stabilizes Mn against dissolution?

Response 4:

We have repeated the related measurement using ICP-MS and found that no evident leaching of Mn ions as the OER proceeds on NaMn₃O₇. As known, the charge disproportionation of Mn ions is the main reason for the leaching of Mn (*Nat. Commun.* **2014**, *5*, 4256; *J. Am. Chem. Soc.* **2013**, *135*, 15670; *J. Am. Chem. Soc.* **2012**, *134*, 1519;). In neutral media, Mn³⁺ is unstable against charge disproportionation and segregates into Mn²⁺ and Mn⁴⁺. The generated Mn²⁺ tends to dissolve into the electrolyte (*Chem. Commun.* **2017**, *53*, 7149). However, this disproportionation reaction becomes unfavorable in alkaline conditions. In this manuscript, the OER were performed in alkaline electrolyte.

In addition to charge disproportionation, the migration of Mn ions in NaMn₃O₇ might lead to dissolution. However, the Mn ions are unlikely to be able to migrate to the interlayer sites for NaMn₃O₇ with the calculated energy barrier as high as 2.80 eV due to the size mismatch between the Mn and Na ions. As such, we attribute the OER durability and structural stability to the ordered native vacancies in NaMn₃O₇ that can self-regulate its deformation and electrochemical reversibility (*Adv. Energy. Mater.* **2018**, *8*, 1800409; *Adv. Energy. Mater.* **2019**, *9*, 1803087.).

Moreover, the XPS spectra of K 2p indicate no intercalation of K⁺ ions into NaMn₃O₇ lattice to replace the Na⁺ site (shown in Supplementary Figure 17). Therefore, the loss of Na⁺ and intercalation/deintercalation of K⁺ will not be responsible for the stability of Mn against dissolution.

Change 4:

1). **Page 11, line 4**, some sentences are revised as follows.

ICP-MS test on the electrolyte further demonstrates that no evident leaching of Na and Mn cations as the OER proceeds on Na_xMn₃O₇ ($x = 1.5, 1$ and 0.7). Actually, Mn ions are unlikely to be able to migrate to the interlayer sites for NaMn₃O₇ with the calculated energy barrier as high as 2.80 eV due to the size mismatch between the Mn and Na ions^{27,38,39}. Moreover, the charge disproportionation reaction that typically leads to the dissolution of Mn ions is unfavorable in alkaline media^{40,41}. As such, we attribute the OER durability and structural stability to the ordered native vacancies in NaMn₃O₇ that can self-regulate its deformation and electrochemical reversibility^{17,27}.

2). **Page 17, line 7**, two literatures are added as follows.

40 Ooka, H., Takashima, T., Yamaguchi, A., Hayashi, T. & Nakamura, R. Element strategy of oxygen evolution electrocatalysis based on in situ spectroelectrochemistry. *Chem. Commun.* **53**, 7149-7161 (2017).

41 Yamaguchi, A. *et al.* Regulating proton-coupled electron transfer for efficient water splitting by manganese oxides at neutral pH. *Nat. Commun.* **5**, 4256 (2014).

Comment 5:

I find it good that the authors quantify activity showing numbers. Unfortunately, stability is not quantified. For instance, the authors write "... the Faradaic efficiency of near-unity is measured for the best-performing catalyst of NaMn_3O_7 , indicating that the measured current is primarily originated from the water oxidation.". This is obviously not enough. 0.9 is also near unity but means a massive loss of material assuming that the rest 10% is due to dissolution. Hence, the exact value must be given and the reason for not having FE=100% must be discussed.

Response 5:

We thank the reviewer for the suggestion. We have revised the related explanation to make it more accurate.

Change 5:

1). Page 10, line 12, one sentence is revised as follows.

In addition, the Faradaic efficiency of 97% is measured for the best-performing catalyst of NaMn_3O_7 , indicating that the measured current is primarily originated from the water oxidation.

Comment 6:

The authors write "negligible loss of specific activity" – how much exactly? In the way how, it shown in Figure 4d, it is difficult to see but it is still visible that the activity drops.

Response 6:

We have revised the related explanation to make it more accurate.

Change 6:

1). Page 10, line 14, two sentences are revised as follows.

The stability measurements were further carried out for $\text{Na}_x\text{Mn}_3\text{O}_7$ at a constant overpotential of 320 mV for 10 h. For NaMn_3O_7 , it maintains 95% of its initial specific activity (Fig. 4d). Similar observations can be made for $\text{Na}_{1.5}\text{Mn}_3\text{O}_7$ and $\text{Na}_{0.7}\text{Mn}_3\text{O}_7$ (Supplementary Fig. 16).

Comment 7:

Stability data should be shown for all materials, not just for one.

Response 7:

We have added the stability data for other three materials as suggested.

Change 7:

1). Page 10, line 14, two sentences are revised as follows.

The stability measurements were further carried out for $\text{Na}_x\text{Mn}_3\text{O}_7$ at a constant overpotential of 320 mV for 10 h. For NaMn_3O_7 , it maintains 95% of its initial specific activity (Fig. 4d). Similar observations can be made for $\text{Na}_{1.5}\text{Mn}_3\text{O}_7$ and $\text{Na}_{0.7}\text{Mn}_3\text{O}_7$ (Supplementary Fig. 16).

2). Supporting information, one figure is added as follows.

Supplementary Fig. 16 | Chronoamperometric curve of $\text{Na}_x\text{Mn}_3\text{O}_7$ at 1.55 V versus RHE.

Comment 8:

Probably a specialist can just this better but for me it seems that Figure S16 does show some amorphization as one would expect based on the ICP-OES data after the XAS experiments (deintercalation without amorphization should be also possible, needs to be discussed).

Response 8:

Thanks for the valuable comment from the reviewer. For NaMn_3O_7 , the Na^+ deintercalation is very difficult from the perspective of thermodynamics with an evident cell volume change of over 20%. (*J. Mater. Chem. A* **2017**, 5, 12752). From XRD patterns (**Supplementary Figure 17**), no evident peak variation was observed for the post-cycled NaMn_3O_7 catalyst after OER measurement. Compared with the reconstructed catalysts reported (*Nat. Energy* **2020**, 5, 881; *Nat. Mater.* **2017**, 16, 925; *Proc. Natl. Acad. Sci.* **2020**, 117, 21906; *Sci. Adv.* **2021**, 7, eabc7323), we conclude that no evident surface amorphization can be found from HRTEM images for the post-cycled catalyst after OER measurement.

Change 8:

1). **Page 10, line 18**, one sentence is revised as follows.

Neither evident surface amorphization in HRTEM images nor peak variation in XRD patterns was observed for the post-cycled catalyst after OER measurement (**Supplementary Fig. 17**).

2). **Page 17, line 16**, one literature is added as follows.

39 Zhang, Z. *et al.* First-principles computational studies on layered $\text{Na}_2\text{Mn}_3\text{O}_7$ as a high-rate cathode material for sodium ion batteries. *J. Mater. Chem. A* **5**, 12752-12756 (2017).

Comment 9:

Assuming that Na dissolves completely from the surface layer, calculate the thickness of the Na-free layer. Why TEM does not show this?

Response 9:

Thanks for the valuable comment from the reviewer. We have checked the structure of $\text{Na}_2\text{Mn}_3\text{O}_7$ after 10 h OER measurement by XRD and HRTEM. As seen, the diffraction peak of the post-cycled $\text{Na}_2\text{Mn}_3\text{O}_7$ exhibits a little shift to a higher angle, confirming the interlayer Na^+ in the lattice is predominantly leached (Supplementary Fig. 19a). Partial surface amorphization with the thickness of 3-5 nm was also observed from HRTEM image for the post-cycled $\text{Na}_2\text{Mn}_3\text{O}_7$ (Supplementary Fig. 19b). Such amorphization is absent in the case of NaMn_3O_7 as discussed in the response to comment 8.

Change 9:

1). Page 11, line 16, one sentence is revised as follows.

Partial surface amorphization with the thickness of 3-5 nm was also observed from HRTEM image for the post-cycled $\text{Na}_2\text{Mn}_3\text{O}_7$ (Supplementary Fig. 19b).

2). Supporting information, one figure is added as follows.

Supplementary Fig. 19 | a, XRD patterns of $\text{Na}_2\text{Mn}_3\text{O}_7$ before and after OER measurement. b, HRTEM of $\text{Na}_2\text{Mn}_3\text{O}_7$ after OER measurement.

Comment 10:

Assuming that there is a gradient in Na⁺ concentration with thickness, how does this change the conclusions from XAS data and mechanistic analysis?

Response 10:

We appreciate the comment from the reviewer. We don't think this would change the conclusions from XAS data and mechanistic analysis. We have confirmed the structural stability of Na_{1.5}Mn₃O₇, NaMn₃O₇ and Na_{0.7}Mn₃O₇ using ICP-MS and XRD pattern. Although some Na⁺ leaching for Na₂Mn₃O₇ occurs during OER, the 2st LSV curve is used for comparison in Fig. 4a. Actually, the activated lattice oxygen from the Na⁺ leaching contributes to the enhanced activity for Na₂Mn₃O₇ during OER stability measurement. This indirect confirms the important role of lattice oxygen oxidation in OER.

Comment 11:

Dissolution data is shown for only one material. Data for other samples should be shown.

Response 11:

We have added the dissolution data for all post-cycled materials after 10 h OER measurement as suggested.

Change 11:

1). **Supporting information**, one table is added as follows.

Supplementary Table 8 | Summary of the results from ICP-MS measurements of the electrolyte collected from stability tests on Na_xMn₃O₇. The reported percentages refer to the mass of the dissolved element with respect to the initial element mass in the electrode.

	Na	Mn
Na ₂ Mn ₃ O ₇	5.10%	0.11%
Na _{1.5} Mn ₃ O ₇	0.23%	0.10%
NaMn ₃ O ₇	0.17%	0.13%
Na _{0.7} Mn ₃ O ₇	0.10%	0.11%

Comment 12:

The authors write “Recent discoveries have demonstrated that the surface metal electrochemical leaching is responsible for partial lattice oxygen loss (Na₂Mn₃O₇ + (1-x/2)H₂O → (2-x)Na⁺ + Na_xMn₃O_{6+x/2} + (2-x)OH⁻)”. According to this equation, Na⁺ dissolves. This must be stated but also discussed.

Response 12:

Thanks for the valuable comment from the reviewer. We have revised the related expressions that may cause confusion and misunderstanding.

Change 12:

1). Page 11, line 17, the original sentence has been deleted and two sentences are added as follows.

We deduce that the high oxidative OER potential drives the Na^+ leaching³⁹. As a result, the activated lattice oxygen from the Na^+ leaching contributes to the enhanced activity for $\text{Na}_2\text{Mn}_3\text{O}_7$.

Comment 13:

Is electrochemical surface area changing with Na^+ leaching? If so, how much and how it influences activity?

Response 13:

Only $\text{Na}_2\text{Mn}_3\text{O}_7$ shows the Na^+ leaching. As the reviewer predicted, the electrochemical surface area of $\text{Na}_2\text{Mn}_3\text{O}_7$ improved 1.68 times after 10 h OER measurement at 1.55 V (shown in Figure SS1 below). However, its OER activity improved 3.22 times after 10 h OER measurement. We believe, the activated lattice oxygen from the Na^+ leaching contributes to the enhanced activity for $\text{Na}_2\text{Mn}_3\text{O}_7$, in addition to the increase of electrochemical surface area.

Figure SS1. Current density differences ($\Delta i/2$) at 1.05 V as a function of scan rate for pristine and post-cycled $\text{Na}_2\text{Mn}_3\text{O}_7$.

Comment 14:

The authors write that “the post-cycled $\text{Na}_2\text{Mn}_3\text{O}_7$ shows the enhanced OER activity than the fresh one (Supplementary Fig. 18d), due to the activation of lattice oxygen from the Na^+ leaching.” This needs to be elaborated more. The main idea of this work that there is an optimal amount of Na. Na=1 is shown to be the most stable. Assuming that Na is leaching from the Na=2 sample, it makes sense that activity is improving. On the other hand, with Na<1, activity is decreasing. Does it mean that eventually this material will lose its activity with more and more Na dissolved?

Response 14:

For $\text{Na}_2\text{Mn}_3\text{O}_7$, some Na^+ can be dissolved into the electrolyte and the molar ratio of Na/Mn decreases. For the best material NaMn_3O_7 , further dissolution of Na^+ during OER measurement is insignificant. To fairly compare the leaching, in the current revised version, we have provided the stability and dissolution data for all materials at 1.55 V for 10 h. ICP-MS (Supplementary Table 8) on the electrolyte further demonstrates that no evident leaching of Na and Mn cations as the OER proceeds on $\text{Na}_x\text{Mn}_3\text{O}_7$ ($x = 1.5, 1$ and 0.7). For

$\text{Na}_2\text{Mn}_3\text{O}_7$, the evident leaching of Na^+ can be observed after OER measurement, whereas the negligible leaching of Mn cations can be observed. As a result, $\text{Na}_x\text{Mn}_3\text{O}_7$ ($x = 1.5, 1$ and 0.7) materials maintain $>94\%$ of their initial specific activity (Supplementary Figure 16).

Reviewer #3

Huang et al. report $\text{Na}_x\text{Mn}_3\text{O}_7$ as a model system to study the mechanism of the oxygen evolution reaction (OER), particularly, the role of oxygen reactivity. The catalysts are studied by DFT, XRD, multi-edge XAS, HRTEM, TEM-EDS, ICP-OES, gas adsorption, gas chromatography, Raman spectroscopy and electrochemical methods. The best catalyst in the series is very active for a manganese oxide and is reasonably stable for 10 h at low overpotential. Mn is also non-toxic and abundant (as is Na). This makes the mechanistic study noteworthy and interesting for the field of electrocatalysis, in particular green hydrogen production. The main claim is that the number of Na^+ correlates with the number of oxygen holes, which seems to be supported by the theoretical calculations (not my area of expertise) but I have doubts about the experimental proofs of the claim (which falls within my area of expertise) as detailed below. Additional evidence and analysis are needed to support the claim(s) as also detailed below. The manuscript is difficult to follow as observations are not clearly separated from interpretations and interpretations are not or too briefly discussed. There are technical issues, particularly with the XAS and electrochemical evaluations. On balance, the topic of lattice oxygen reactivity and the materials system are interesting and the role of alkali ions clearly deserves more attention. However, many additional experiments and discussion are needed to strengthen the manuscript for publication. Please address:

Comment 1:

Claims. **a)** The presented data do not clearly support the involvement of lattice oxygen. The in situ XAS do not support that the Mn valence remains unchanged. Are changes expected in the in situ XAS for rods of 200 nm width and 800 nm length (Suppl. Fig. 13)? How much does the surface contribute to the signal? I doubt that Mn-K edge XAS is suited to resolve surface changes, which is also supported by the clearer changes in soft X-ray XAS in Fig. 3. A more surface-sensitive method, e.g. soft XAS or XPS is required. **b)** I am wondering how controlled the active state is when Na leaches from the as-synthesized material. Do all leach Na and take up H^+ ? The elucidation of the composition of all catalysts is needed to support that the active state can be rationally designed.

Response 1:

Thanks for the reviewer's critical and inspiring comments. We provide our itemized response below.

Response 1a: It is true that in *situ* XAS measurement could reflect the change of coordination environment of metal centre from both bulk and surface. Currently, many literatures have reported that the use of in situ XAS measurement can resolve the near-surface structures under electrochemical condition of the catalyst (*Nat. Catal.* **2020**, *3*, 743; *Nat. Catal.* **2018**, *1*, 711; *Nat. Catal.* **2018**, *1*, 841; *Nat. Energy* **2018**, *3*, 140; *Nat.*

Mater. **2017**, *16*, 925). Absolutely, in situ soft XAS or XPS technique is more accurate and better suited to resolve surface changes. Therefore, we have supplemented the XPS measurement of catalyst working at various applied potentials (Supplementary Fig. 20). As seen, the oxidation state of surface Mn increases in NaMn_3O_7 when the potential is increased from OCV to 1.25 V. Further increasing potential to 1.55 V, there is no evident variation of Mn oxidation state, in agreement with the results from in situ XAS measurement. Therefore, we conclude that OH^- (aq.) can refill the oxygen vacancy sites of NaMn_3O_7 under electrochemical condition, thus the electrophilic oxygen species with oxygen hole can be formed after the subsequent deprotonation and involve in the subsequent O-O bond formation.

Response 1b: To fairly compare the leaching, in the current revision, we have provided the stability and dissolution data for all materials at 1.55 V for 10 h. For $\text{Na}_{1.5}\text{Mn}_3\text{O}_7$, NaMn_3O_7 and $\text{Na}_{0.7}\text{Mn}_3\text{O}_7$ materials, insignificant dissolution of Na^+ ions is observed under electrochemical OER condition. But for $\text{Na}_2\text{Mn}_3\text{O}_7$, partial dissolution of Na^+ into the electrolyte can occur. We think the removal of Na^+ ions for $\text{Na}_2\text{Mn}_3\text{O}_7$ depends on the potential applied to the electrode. For $\text{Na}_{1.5}\text{Mn}_3\text{O}_7$, NaMn_3O_7 and $\text{Na}_{0.7}\text{Mn}_3\text{O}_7$ materials, the applied potential during OER lower than 1.8 V is not enough to further remove the Na^+ ions in lattice (*arXiv preprint arXiv* **2020**, 2010.13107; *Adv. Energy. Mater.* **2019**, *9*, 1803087).

We also performed DFT calculations to confirm that the H^+ cannot occupy the Na vacancy. As seen in Figure SS2a-c, when H is replaced by Na, the H can easily migrate on the neighboring O2 site (O2: coordinated with two Mn ions). More importantly, the simulated XRD pattern (Figure SS2d) shows the evident peak shift after H incorporation into NaMn_3O_7 . However, in experimental, no evident peak variation in XRD patterns was observed for the post-cycled NaMn_3O_7 catalyst after OER measurement (Supplementary Fig. 17). Therefore, we believe H^+ cannot occupy the Na vacancy in the composition of all catalysts.

Figure SS2. (a-c) The structural optimization of $\text{Na}_{1.5}\text{Mn}_3\text{O}_7$, NaMn_3O_7 , $\text{Na}_{0.5}\text{Mn}_3\text{O}_7$ after H-incorporation to occupy the Na vacancy. (d) The simulated XRD pattern of NaMn_3O_7 after H-incorporation.

Change 1:

1). Page 11, line 30, three sentences about the analysis of in situ Mn 2p XPS spectra are added as follows.

This is further corroborated by X-ray photoelectron spectroscopy (XPS) measurements, which revealed the change in electronic structure at the surface of the catalysts working at various applied potentials (Supplementary Fig. 20). As seen, the oxidation state of surface Mn increases in NaMn_3O_7 when the potential is increased from OCV to 1.25 V. There is no evident variation of Mn oxidation state with further increasing potential to 1.55 V.

2). Supporting information, one figure is added as follows.

Supplementary Fig. 20 | XPS spectra of Mn 2p of NaMn_3O_7 under open circuit, 1.25 and 1.55 V (versus RHE).

3). Supporting information, the missing experimental details about in situ XAS are added below.

In situ XAS measurements were performed in fluorescence mode using a home-made electrochemical cell. The catalysts were sprayed on carbon paper at a loading of 2 mg cm^{-2} as the working electrode.

4). Supporting information, the experimental details about XPS working at various applied potentials are added below.

X-ray photoelectron spectroscopy (XPS) measurements were performed using a Thermo ESCALAB 250Xi X-ray photoelectron spectroscope. The home-made X-ray cell is composed of three chambers: a reaction chamber, an analysis chamber and a preparation chamber. The back side of the Si_3N_4 window with deposited catalyst on Au/Ti layer faced into the interior of the electrochemical cell. At different applied potentials, the working electrodes were first polarized for 5 min to reach a steady state, then the corresponding XPS signals were collected and analyzed.

Comment 2:

Please separate observations from interpretation and provide additional information so that the interpretations can be followed by non-experts. **a)** What is the significance of the magnetization moments of the oxygen ions? Why is it important for catalysis? What is the relation to the parameter P1? **b)** How are the parameters P1 and P2 calculated? Why has ideal catalysis the values in Suppl. Table 2? I would have

assumed they are zero? **c)** Why should Na leaching activate lattice oxygen? It needs more discussion and is an interpretation. **d)** What is the conclusion of the Raman experiments? It needs discussion. Does it indicate that TMAOH intercalates?

Response 2:

We have separated observations from interpretation and provided additional information so that the interpretations can be followed by non-experts.

Response 2a: Magnetization moment results from the presence of unpaired electrons in materials, and the fully paired electrons lead to a net magnetic moment of zero. For $\text{Na}_2\text{Mn}_3\text{O}_7$, the oxygen ions tend to form 8-electron stable structures with no unpaired electrons. When the number of Na^+ decreases from $x=2$, lattice oxygen ions are activated and oxidized, meanwhile Mn would be in their maximum achievable oxidation states of 4+. Thus, the unpaired electrons are generated in such oxidized oxygen ions. To quantify the number of unpaired electrons, the concept of ligand (oxygen) hole is proposed in this manuscript.

Activating lattice oxygen to generate spin-characteristic ligand holes are capable of tuning the lattice oxygen reactivity that links to energy barrier symmetry between O-H bond cleavage and $^*\text{OOH}$ formation. As a result, the minimum theoretical overpotential of ~ 0.4 eV can be achieved when the number of oxygen holes is optimized to the top of a volcano plot (Fig. 2d).

P1 is defined as the variation of $\Delta G^*_{\text{O}} - \Delta G^*_{\text{OH}}$ with reference to S1. As seen, the greater number of oxygen hole, the higher value of P1. The minimum theoretical overpotential is ultimately approached when the magnetization moment of oxygen ions increases above $0.23 \mu\text{B}$.

Response 2b: The number of Na ions regulates both electronic effect (P1) and geometric effect (P2) to determine the overall adsorption energy of oxygen-containing intermediates ($\Delta G^*_{\text{OH}} = \Delta G^*_{\text{OH}} + \text{P1}$; $\Delta G^*_{\text{O}} = \Delta G^*_{\text{O}} + 2\text{P1}$; $\Delta G^*_{\text{OOH}} = \Delta G^*_{\text{OOH}} + \text{P1} + \text{P2}$). The electronic parameter of P1 is defined as the variation of $\Delta G^*_{\text{O}} - \Delta G^*_{\text{OH}}$ with reference to S1, where the oxygen ions in S1 have the fully paired electrons with a net magnetic moment of zero. The geometric parameter (P2) is defined as the variation of ΔG^*_{OOH} with reference to S6, where the noncovalent interaction between alkali metal and $^*\text{OOH}$ in S6 is zero due to the absence of Na^+ .

In Suppl. Table 2, an ideal OER catalyst is that the free energy change of all OER steps are numerically equal to the equilibrium potential of 1.23 eV. According to the definition of P1 and P2, we can invert the values of P1 (0.56 eV) and P2 (-0.78 eV) for catalysts with overpotential of zero. However, the number of Na ions regulates both P1 and P2 and such two parameters cannot be optimized independently. Therefore, we cannot really achieve one catalyst with overpotential of zero by tuning the number of Na ions.

Response 2c: For $\text{Na}_2\text{Mn}_3\text{O}_7$, the oxygen ions tend to form 8-electron stable structures with no unpaired electrons. According to the amplitude of charge transfer energy (Δ) and $d-d$ Coulomb interaction (U), $\text{Na}_2\text{Mn}_3\text{O}_7$ ($U > \Delta$) is located at charge-transfer regime, showing an empty metallic band lying above the fully filled $|\text{O}_{2p}$ band (Fig. 1a). For charge balance, when the number of Na^+ decreases from $x=2$ to $x=0.5$, the electrons are removed from fully filled $|\text{O}_{2p}$ band due to its higher energy than metal d-band. As a result,

the lattice oxygen ions are activated and oxidized, meanwhile Mn would be in their maximum achievable oxidation states of 4+.

Response 2d: Raman experiments are performed to confirm the formation of negative-charged oxygen-containing intermediates during OER. Tetramethylammonium cation (TMA^+) is used as a chemical probe because of its specific electrostatic adsorption interaction with negative oxygenated species, not the intercalation of TMAOH into NaMn_3O_7 (*Nat. Energy* **2019**, 4, 329; *Angew. Chem. Int. Ed.* **2017**, 56, 8652; *J. Am. Chem. Soc.* **2015**, 137, 15112).

Change 2:

1). Page 3, line 28, one sentence is added below.

More details about calculating the number of oxygen hole via crystal field theory, magnetization moment and bader charge can be found in Supplementary Fig. 1.

2). Page 6, line 1, one figure is revised below.

Fig. 2 | Constructing better OER electrocatalyst through tuning the lattice oxygen reactivity and scaling relation.

a, Scheme of rational design of better $\text{Na}_x\text{Mn}_3\text{O}_7$ electrocatalysts. **b**, Shifts in P1 to regulate the theoretical overpotential (η) by tuning the magnetization moment of O. **c**, Shifts in P2 to reduce the overpotential ceiling (η_{min}) by tuning Na^+ -specific noncovalent interaction to overcome the LSR. **d**,

Dynamic volcano plot (η versus $\Delta G_{*O} - \Delta G_{*OH}$) derived from the rebuilt LSR. **e**, Unified volcano plot (η versus P1-P2) using a binary descriptor of P1-P2.

3). Page 13, line 25, some sentences are revised below.

To track these charged intermediates on NaMn_3O_7 during OER, tetramethylammonium cation (TMA^+) as a chemical probe is introduced to the solution because of its specific electrostatic interaction with negative oxygenated intermediates^{8,47}. As expected from the Raman spectra (Fig. 5f), there are three new peaks appear at 451, 753 and 951 cm^{-1} , coinciding with the characteristic peaks of TMA^+ , when the NaMn_3O_7 electrode was operated at a constant potential of 1.50 V versus RHE in 1 M tetramethylammonium hydroxide (TMAOH) electrolyte. We further compare the OER activities of NaMn_3O_7 in 1 M KOH and TMAOH solutions (Supplementary Fig. 26). There is decreased OER activity with the change of Tafel slope from 48.3 to 58.1 mV dec^{-1} because of the partial inhibition of the OER, as a result of strong electrostatic interaction between TMA^+ and negative oxygenated intermediates.

4). Supporting information, one figure is added below.

Supplementary Fig. 26 | Comparison of polarization curves (**a**) and derived Tafel slopes (**b**) of NaMn_3O_7 in 1 M KOH and 1 M TMAOH ($\text{OI}^{\delta-}$ represents the negative oxygenated intermediates.).

Comment 3:

XAS analysis. **a)** Fig. 3c does not support the interpretation of its FT in Fig. 3d. It appears that $x=2$ and $x=1$ have very similar wavenumbers where $y=0$ (meaning similar distance) and similar amplitude for low wavenumber (meaning similar coordination number of Mn-O). The discussion of interatomic distances and coordination numbers requires EXAFS fits., preferably in k -space. **b)** The x -axis labels in Fig. 3d and Fig. 5b are misspelled. It is not a “radical” but a “radial” distance. Actually, reduced distance would be most appropriate as there is not mentioning that the authors corrected the axis of the phase shifts. **c)** why are only $x=2$ and $x=1$ shown in Fig. 3d,e,f.? What about the other samples? **d)** Fig. 3e does not show the O-K main edge. No conclusions can be made about it. The pre-edge is shown. Please discuss why the shift is fully assigned to the oxygen valence and not to changes of the Mn ligand field. Are there other reports in literature where oxygen holes are studied by the O K pre-edge? **e)** Three reference materials are insufficient

for a good calibration curve for the Mn valence. Furthermore, the edge position depends on the type of ligand. Mn as the ligand of the 1st coordination shell is drastically different as compared to O. **f)** The Mn-O bond distance of 1.2 Å is much too short. This is the reduced distance. Even when the phase shifts are accounted for, the distance is still too short. Please compare it to the Mn-O distances from the $\text{Na}_x\text{Mn}_3\text{O}_7$ crystal structure. Perhaps an EXAFS fit gives a more realistic value. **g)** Please quantify the “hybridization feature” in Fig. 3e. Can the change be explained by Mn oxidation?

Response 3:

We appreciate the reviewer’s critical and inspiring comment and provide our itemized response below.

Response 3a: We have fitted Mn K-edge EXAFS curves and the corresponding results are showed in Supplementary Table 7. As the number of Na^+ is reduced, the coordination number of Mn-O decreases while the interatomic distance of Mn-O increases (Supplementary Table 7), further suggesting the presence of oxygen vacancy and Mn^{3+} ions in bulk.

Response 3b: We have corrected the mistakes as suggested by the reviewer.

Response 3c, d: NaMn_3O_7 is the best material in our work, as a result of the balance between tuning of lattice oxygen reactivity and scaling relation via alkali metal mediation. To confirm the enhanced lattice oxygen activity of NaMn_3O_7 , we use $\text{Na}_2\text{Mn}_3\text{O}_7$ as an example for comparison (The choice of other samples like $\text{Na}_{1.5}\text{Mn}_3\text{O}_7$ will serve the same purpose) and measured their O K-edge spectra, as the OER activity for $\text{Na}_2\text{Mn}_3\text{O}_7$ is constrained by lattice oxygen reactivity with the rate-limiting step of *OOH formation and it will be able to demonstrate the difference. As seen in Fig. 3e, the characteristic peaks between 528 and 534 eV represent the spectroscopic excitations to the hybridized state of O-2p and Mn-3d, which are split by the crystal field of the local Mn-O coordination geometry. Moreover, the evident shift of the O K-edge to a higher energy region confirms the appearance of oxygen hole states in $|\text{O}_{2p}$ (that is, lattice oxygen oxidation), which agrees well with the reported literatures (*Nat. Commun.* **2020**, *11*, 4973; *Nat. Mater.* **2019**, *18*, 256).

Response 3e: It is true that three reference materials are insufficient for a good calibration curve for the Mn valence. Thus, we have deleted the related part to make it more accurate.

Response 3f: It is true that the radial distance shown in Fig. 3d is the reduced distance without correcting the axis of the phase shifts. In Supplementary Table 7, the fitted results provide a more realistic value.

Response 3g: We have quantified the “hybridization feature” in Fig. 3e. The hybridization of Mn-O bonds of

$\text{Na}_2\text{Mn}_3\text{O}_7$ and NaMn_3O_7 can be assessed by the integrated intensities of the pre-edge region normalized to oxygen content per formula unit and the nominal number of empty Mn3d states in both e_g and t_{2g} symmetry (*Nat. Commun.* **2013**, *4*, 2439; *J. Phys. Chem. C* **2014**, *118*, 1856).

Change 3:

1). **Page 9, line 6**, two sentences are revised below.

A clear loss of intensity on Mn 3d-O 2p hybridization feature is observed when the number of Na^+ decreases, implying a decrease of Mn oxidation state on the surface, in line with the results from the Mn K-edge

XANES spectra (Fig. 3b). The hybridization parameters (defined as absorbance/(e_g holes + $1/4t_{2g}$ holes)) of Mn-O bonds of $\text{Na}_2\text{Mn}_3\text{O}_7$ and NaMn_3O_7 are calculated to be about 0.45 and 0.51³³.

2). Page 16, line 9, three literatures are added below.

20 Hong, J. *et al.* Metal-oxygen decoordination stabilizes anion redox in Li-rich oxides. *Nat. Mater.* **18**, 256-265 (2019).

33 Grimaud, A. *et al.* Double perovskites as a family of highly active catalysts for oxygen evolution in alkaline solution. *Nat. Commun.* **4**, 2439 (2013).

34 Ning, F. *et al.* Inhibition of oxygen dimerization by local symmetry tuning in Li-rich layered oxides for improved stability. *Nat. Commun.* **11**, 4973 (2020).

3). Page 8, line 16, one sentence is revised as follows.

As the number of Na^+ is reduced, the coordination number of Mn-O decreases while the interatomic distance of Mn-O increases (Supplementary Table 7), further suggesting the presence of oxygen vacancy and Mn^{3+} ions in the bulk³⁰.

4). Supporting information, the fitting results of Mn K-edge EXAFS curves are shown below.

Supplementary Table 7 | Fitting results of Mn K-edge EXAFS curves. (^aR: bond distance; ^bCN: coordination numbers; ^c σ^2 : Debye-Waller factors; ^d S_0^2 : Amplitude attenuation factors.)

Sample	bond	^a R (Å)	^b CN	^c σ^2 (Å ²)	^d R factor (%)
$\text{Na}_2\text{Mn}_3\text{O}_7$	Mn-O	1.907	6.0 (fixed)	0.005	1.67
$\text{Na}_{1.5}\text{Mn}_3\text{O}_7$	Mn-O	1.910	5.78	0.004	1.20
NaMn_3O_7	Mn-O	1.913	5.31	0.003	1.89
$\text{Na}_{0.7}\text{Mn}_3\text{O}_7$	Mn-O	1.910	5.10	0.003	1.78

Comment 4:

Analysis of the Mn and O valences. **a)** Can it be excluded that proton intercalation occurs rather than oxygen oxidation? If so how? **b)** The formulation “This counterintuitive phenomenon could be rationalized by the possible lattice oxygen losses due to the increased Mn-O covalency” Is misleading as it is later discussed that both the number of O and their valence may change.

Response 4:

Response 4a: Please refer to discussion in **Response 1b**. We demonstrate that H^+ cannot occupy the Na vacancy in the composition of all catalysts.

Response 4b: We have revised the discussion about Mn-O covalency.

Change 4:

1). Page 8, line 10, one sentence is revised below.

This indicates that Mn oxidation state in $\text{Na}_x\text{Mn}_3\text{O}_7$ is lowered with the presence of oxygen vacancy as the number of Na^+ decreases²⁸.

Comment 5:

Introduction.

Catalytic mechanisms based on lattice oxygen should be mentioned explicitly in the introduction. Both nucleophilic attack and O-O coupling are possible without the involvement of lattice oxygen.

Response 5:

Thanks for the reviewer's comment. We have added the discussion about catalytic mechanisms based on lattice oxygen explicitly in the introduction.

Change 5:

1). Page 2, line 4, two sentences are added below.

Considering the origin of O_2 product, there are two widely accepted OER mechanisms including adsorbate evolution mechanism (AEM) and lattice oxygen oxidation mechanism (LOM)^{5,7}. Regardless of which OER mechanism is applicable on a catalyst surface, it has been reported that O-O bond formation can follow two different pathways, i.e., acid-base nucleophilic attack and O-O direct coupling^{7,8}.

Comment 6:

Lattice oxygen must be defined. Suppl. Fig. 1 shows (a) μ 3-oxo, (b) μ 2-oxo, (c) μ 1-oxo motifs. Only (a) and (b) are lattice oxygen.

Response 6:

We appreciate the reviewer's comment. We worried that using these two terms (μ 3-oxo and μ 2-oxo) will cause misunderstanding, because the coordinated metals include Mn and Na in different numbers. Therefore, we have defined the lattice oxygen using O1 and O2, where O1 is coordinated with three Mn ions and O2 is coordinated with two Mn ions, respectively. Such definition allows us to demonstrate more clearly the formation of oxygen holes in oxygen lone pair states ($|\text{O}_{2p}$).

Comment 7:

DFT calculations. **a)** Which motif was the active site in the DFT calculations, μ -3-oxo (Suppl. Fig. 1a) or μ -2-oxo (Suppl. Fig. 1b)? **b)** How would it change the conclusions of the DFT results if Na^+ was partially replaced by H^+ as suggested by the experimental results? **c)** Are the clusters sufficiently large to exclude size effects, e.g. due to mirror charges?

Response 7: We appreciate the reviewer's comments and provide our itemized response below.

Response 7a: In the DFT calculations, lattice oxygen of μ -2-oxo is coordinated with two Mn ions (that is, O2) as the active site. One of the $\text{O}(2p)$ orbitals pointing towards Mn vacancy in μ -2-oxo coordination environment is non-bonded. As the number of Na^+ decreases, these high energy non-bonded oxygen states are activated and lattice oxygen ions are oxidized. The activated oxygen ions can function as active sites to promote the O-O bond formation via either acid-base nucleophilic attack or O-O direct coupling.

Response 7b: More discussion can be found in **Response 1b**. We demonstrate that H^+ cannot occupy the Na vacancy in the composition of all catalysts. According to Pourbaix diagram (**Supplementary Fig. 23**), the surface termination of $Na_xMn_3O_7$ should be the stoichiometric composition under electrochemical OER conditions. Therefore, the used models in our calculation are reasonable to predict the activity in experiments.

Response 7c: Taking $NaMn_3O_7$ as example, we have compared the elementary reaction free energy for deprotonation of *O and formation of *OOH (usually the potential-determining step in OER) on 1×1 , 1×2 and 2×2 $NaMn_3O_7$ supercell. It is found that the results are very close to the previous one (1×1 supercell). Therefore, we believe, the current clusters used are sufficiently large to exclude size effects, e.g. due to mirror charges.

Change 7:

1). Computational models and methods, one sentence is added below.

The size of such supercells is suitable for accurate calculation of surface OER reaction (**Supplementary Table 9**).

2. Supporting information, one table is added below.

Supplementary Table 9 | Comparison of elementary reaction free energy for deprotonation of *O and formation of *OOH on 1×1 , 1×2 and 2×2 $NaMn_3O_7$ supercell. The overpotential of potential-determining step is also shown.

Model	ΔG_2 (eV)	ΔG_3 (eV)	η (eV)
1×1	1.556	1.544	0.326
1×2	1.555	1.543	0.325
2×2	1.588	1.509	0.358

Comment 8:

Descriptor P1-P2. **a)** how transferable is the descriptor P1-P2 to other materials classes/families? **b)** do both P1 and P2 scale linearly with Na? **c)** why is P1 positive and P2 negative?

Response 8:

We appreciate the reviewer' inspiring comments and provide our response below.

Response 8a: In this manuscript, $Na_xMn_3O_7$ ($0 < x \leq 2$) materials, as one type of alkali metal-incorporated metal oxide, provide a good platform for unveiling how to rationally design better OER electrocatalysts through tuning lattice oxygen reactivity and scaling relation mediated by alkali metal ion. We believe the descriptor P1-P2 can be transferable to many other alkali metal-incorporated metal oxides such as $Li(Na)_xCoO_2$ ($0 < x \leq 1$) and Li_xIrO_2 ($0 < x \leq 1$). Currently, many literatures have reported that such materials exhibit superior activity upon removing part of Li^+ (*Nat. Commun.* **2020**, *11*, 1984; *J. Am. Chem. Soc.* **2019**, *141*, 3014; *Proc. Natl. Acad. Sci.* **2019**, *116*, 23473; *Nat. Energy* **2016**, *2*, 16189).

Response 8b: The values of P1 and P2 are determined by the number of Na^+ , but are not scaled linearly (see Figure SS3 below).

Figure SS3. Correlation between the parameters of P1 and P2 and the number of Na⁺ in Na_xMn₃O₇.

Response 8c: The adsorption energy of oxygen-containing intermediates are determined according to: $\Delta G'^{*}_{OH} = \Delta G^{*}_{OH} + P1$; $\Delta G'^{*}_O = \Delta G^{*}_O + 2P1$; $\Delta G'^{*}_{OOH} = \Delta G^{*}_{OOH} + P1 + P2$. P1 is defined as the variation of $\Delta G^{*}_O - \Delta G^{*}_{OH}$ with reference to S1. Adsorption of oxygen-containing intermediates is hindered as the number of Na⁺ reduces, because of the increased number of oxygen holes in |O_{2p} upon activating lattice oxygen. Therefore, the P1 shows the positive value. P2 is defined as the variation of ΔG^{*}_{OOH} with reference to S6, where the noncovalent interaction between alkali metal and *OOH in S6 is zero due to the absence of Na⁺. Na⁺-specific interaction on *OOH is promoted as the number of Na⁺ increases. Therefore, the P2 shows the negative value.

Comment 9:

Why is the reaction order fractional? Was it ensured that the intercalation of H⁺ ceased before the catalytic investigation?

Response 9:

When referred to RHE, the thermodynamic driving force of the over-all PCET reaction does not depend on pH. The reaction order would then be zero if the reaction follows the concerted pathway, as the proton is never decoupled from the electron in this case. On the other hand, the sequential pathway, when limited by the proton concentration, gives a non-vanishing reaction order in pH on the RHE scale. Fractional reaction orders are typically expected when the mechanism consists of complex chain reactions or when there is a competition between direct product formation and side reactions. Side reactions can be due to the presence of unstable species or secondary phases or their generation during the reaction (*Catal. Today* **2016**, 262, 2). Actually, it is a common phenomenon that the reaction order is fractional as reported in many related OER literatures (*Nat. Commun.* **2019**, 10, 5074; *Nat. Chem.* **2017**, 9, 457; *ACS Energy Lett.* **2018**, 3, 2884; *Catal. Today* **2016**, 262, 2). The redox of active sites would occur before the occurrence of OER. As a result, the apparent reaction order of H⁺ is calculated to be fractional. In essence, the pH-dependent activity demonstrates the presence of non-concerted proton-electron transfer processes in the rate-determining step.

More discussion about the intercalation of H^+ can be found in **Response 1b**. We demonstrate that H^+ cannot occupy the Na vacancy in the composition of all catalysts. Meanwhile, OH^- (aq.) tends to spontaneously refill the oxygen vacancy sites of $NaMn_3O_7$ under 1.23 V vs. RHE. According to Pourbaix diagram (**Supplementary Fig. 23**), the surface termination of $NaMn_3O_7$ should be the stoichiometric composition upon subsequent deprotonation under electrochemical OER conditions.

Change 9:

1). **Page 13, line 14**, two sentences are revised below.

From pH-dependent OER measurements on RHE scale, $NaMn_3O_7$ shows the enhanced activity with the increase of pH from 12.5 to 14. The strong pH dependence indicates the chemical deprotonation step is rate-limiting (**Fig. 5d**).

Comment 10:

Minor. **a)** why evaluation at $0.25 \text{ mA/cm}^2_{ox}$. Is it also used elsewhere? **b)** Fig. 5c is not clear. It looks like a terminal O is healed on the right but it is a μ -2-oxo bridge on the left. **c)** Fig. 5e is not clear. On which oxygen sites does OH adsorb? A μ -2-oxo site? If so, no $\sim 180^\circ$ M-O-M angle should be drawn.

Response 10: We appreciate the reviewer's comments and provide our response below.

Response 10a: In this manuscript, the currents are normalized by Brunauer-Emmett-Teller (BET) surface areas to reflect the intrinsic activity of the catalysts. We follow the choice of $0.25 \text{ mA/cm}^2_{ox}$ as it is also used in many reported literatures (*Nat. Catal.* **2020**, 3, 554; *Nat. Catal.* **2020**, 3, 516; *Chem. Soc. Rev.* **2019**, 48, 2518).

Response 10b: For $Na_xMn_3O_7$ slabs, they contained two and three coordinated O with Mn ions. No terminal oxygen can be found. To make it clearer, we have revised **Fig. 5c** using the $NaMn_3O_7$ slab.

Response 10c: We agree with the reviewer and have revised **Fig. 5e** accordingly.

Change 10:

1). **Page 12, line 18**, Fig. 5 is revised below.

Fig. 5 | Evidence of lattice oxygen as reaction site under OER conditions. **a,b**, Normalized *in situ* Mn K-edge XANES spectra (**a**) and Fourier transformed EXAFS $k^3\chi(k)$ oscillation functions (**b**) of NaMn_3O_7 under open circuit, 1.25 and 1.55 V (versus RHE). **c**, The process of oxygen vacancy refilling under 1.23 V and the associated energy barrier. **d**, pH dependence of the OER activities of NaMn_3O_7 . The inset is the proton reaction order estimated by $\rho^{\text{RHE}} = (\partial \log(i) / \partial \text{pH})$, with ρ^{RHE} and i being the proton reaction order and current density at a fixed potential of 1.55 V vs. RHE. **e**, Schematic OER pathway of acid-base nucleophilic attack involving the rate-limiting proton transfer decoupled from electron transfer step. The inset illustrates the formation of negatively charged oxide surface when equilibrated with the electrolyte. **f**, Raman spectra of NaMn_3O_7 electrodes. These electrodes were respectively operated at 1.55 V versus RHE in 1 M KOH (black line) and TMAOH (red line) solution, then were thoroughly washed by rinsing with high-purity water and acetone for *ex situ* Raman spectroscopy measurement.

REVIEWER COMMENTS

Reviewer #1 (Remarks to the Author):

All my concerns have been addressed and the paper is now publishable.

Reviewer #2 (Remarks to the Author):

Thank you for addressing my concerns thoroughly, I can now recommend the publication of the manuscript. The only minor revision needed – the authors should provide experimental details for ICP-MS data (calibration concentrations, internal standards, etc.).

Reviewer #3 (Remarks to the Author):

Huang et al. have added additional discussion and analyses to clarify questions and concerns of all reviewers. Unfortunately, many of my comments have not been addressed fully. In particular, the XAS data are likely overinterpreted. The new XPS data seems to support the claims better and should be highlighted instead of XAS. The point of the oxygen vacancies should be strengthened by further revision of the otherwise coherent and convincing manuscript.

1a. Are changes expected in the in situ XAS for rods of 200 nm width and 800 nm length (Suppl. Fig. 13)? What volume fraction is changed? Is the expected change detectable by TM K edge XAS? It is not very helpful to refer to other publications (unless they performed the exact same experiment with the same material and setup). A related question that I did not ask explicitly: are the vacancies created in the bulk or only on the surface?

By how much does the edge shift? It seems that it is a very small shift.

Suppl. Fig. 20. By convention, the x axis of XPS is plotted with the highest energy on the left. The oxidation of the material is an interpretation, not an observation. It is not stated on what observation the interpretation is based. If in situ XPS can be performed: why wasn't the O 2p binding studied, which gives more direct insight into oxygen vacancy formation?

Fig. 5a does not support the statement "indicating the oxygen vacancy in NaMn₃O₇ can be healed with OH⁻ before the electrochemical OER process". There is no clear change.

Fig. 5b does not support the statement "the amplitude of Mn-O peak increases while the position of Mn-O peak shift towards lower value from OCV to 1.25 V". The change is within the noise level and not supported by an EXAFS fit.

Suppl. Fig. 20 support that there is oxidation from OCV to 1.25 and no further oxidation to 1.55 V. It should be shown in the main text and Fig. 5a and 5b (re)moved. Why should this Mn oxidation be coupled to oxygen vacancy healing and not a deprotonation on the surface, which is commonly proposed?

2a-c. The replies to the reviewers is clear. Please add it to the manuscript or Supplemental information so that the readers can follow your arguments

3a. Thanks for performing EXAFS simulations. Please perform them with identical sigma² (in addition to the free fits) to support reduction of the coordination numbers CN. CN and sigma² correlate strongly. It is suspicious that both fit variables are reduced. Shouldn't R increase linearly with the reducing the Na stoichiometry?

3d. Fig. 3e does not show the O-K main edge. No conclusions can be made about it. The pre-edge is shown. Please update the text accordingly.

Please discuss for the readers why the shift of the pre-edge onset is fully assigned to the oxygen valence and not to changes of the Mn orbitals as is discussed on P10L9

Are there other reports in literature where oxygen holes are studied by the OK pre-edge?

Nat. Commun. 2020, 11, 4973 -> cannot find it. A typo?

Nat. Mater. 2019, 18, 256) -> clearly states "O K-edge sXAS and RIXS further support pure multivalent, hybridized Ir-O redox in the absence of oxygen redox in LIO."

I find the interpretation of the O-K pre-edge is still insufficiently discussed or unsupported by the data.

6. Please add the definition of "lattice oxygen" to the figure or its caption.

8b. Fig. S3 should be added as supplemental figure as it is important to understand the connection between P1 and P2 as well as the limits for materials optimization using Na stoichiometry.

The documentation of the XAS experiments is still insufficient. What detector was used for the fluorescence yield? How was the synchrotron beam monochromatized? What beamline was used? Details of the Fourier transform are missing. The work is not reproducible when the energy/k-range and window function of the Fourier transform is omitted.

We thank the reviewers for their time and very useful comments in improving the quality of this manuscript. Provided below is our detailed point-to-point response to each question. The changes in the manuscript have been highlighted and listed below.

Reviewer #1

All my concerns have been addressed and the paper is now publishable.

Response #1:

We thank the reviewer for the valuable suggestions in improving the quality of this manuscript.

Reviewer #2

Thank you for addressing my concerns thoroughly, I can now recommend the publication of the manuscript.

Comment 1:

The only minor revision needed – the authors should provide experimental details for ICP-MS data (calibration concentrations, internal standards, etc.).

Response 1:

We thank the reviewer for the suggestion. In the current revision, we have added the experimental details for ICP-MS data (calibration concentrations, internal standards, etc.).

Change 1:

1). Supporting information, the experimental details for ICP-MS data are added below.

For the ICP-MS measurement, the standard and blanks were prepared in trace metals grade 2% HNO₃ diluted in milli-Q water. The standard curve included concentrations 0.1, 1.0, 10.0, 100, 500, and 1000 ppb and was linear over this range. All sample concentrations fell within the concentration range described by the calibration curves. The internal standard (10 µg L⁻¹ of Rh) was used.

Reviewer #3

Huang et al. have added additional discussion and analyses to clarify questions and concerns of all reviewers. Unfortunately, many of my comments have not been addressed fully. In particular, the XAS data are likely overinterpreted. The new XPS data seems to support the claims better and should be highlighted instead of XAS. The point of the oxygen vacancies should be strengthened by further revision of the otherwise coherent and convincing manuscript.

Comment 1:

Are changes expected in the in situ XAS for rods of 200 nm width and 800 nm length (Suppl. Fig. 13)? What volume fraction is changed? Is the expected change detectable by TM K edge XAS? It is not very helpful to refer to other publications (unless they performed the exact same experiment with the same material and setup). A related question that I did not ask explicitly: are the vacancies created in the bulk or only on the surface? By how much does the edge shift? It seems that it is a very small shift. By convention,

the x axis of XPS is plotted with the highest energy on the left. The oxidation of the material is an interpretation, not an observation. It is not stated on what observation the interpretation is based. If in situ XPS can be performed: why wasn't the O 2p binding studied, which gives more direct insight into oxygen vacancy formation? Fig. 5a does not support the statement "indicating the oxygen vacancy in NaMn_3O_7 can be healed with OH^- before the electrochemical OER process". There is no clear change. Fig. 5b does not support the statement "the amplitude of Mn-O peak increases while the position of Mn-O peak shift towards lower value from OCV to 1.25 V". The change is within the noise level and not supported by an EXAFS fit. Suppl. Fig. 20 support that there is oxidation from OCV to 1.25 and no further oxidation to 1.55 V. It should be shown in the main text and Fig. 5a and 5b (re)moved. Why should this Mn oxidation be coupled to oxygen vacancy healing and not a deprotonation on the surface, which is commonly proposed?

Response 1:

Thanks for the reviewer' critical and inspiring comments. We provide our itemized response below.

Comment 1a:

Are changes expected in the in situ XAS for rods of 200 nm width and 800 nm length (Suppl. Fig. 13)? What volume fraction is changed? Is the expected change detectable by TM K edge XAS? It is not very helpful to refer to other publications (unless they performed the exact same experiment with the same material and setup). A related question that I did not ask explicitly: are the vacancies created in the bulk or only on the surface? By how much does the edge shift? It seems that it is a very small shift.

Response 1a:

Thanks for the reviewer' critical comments. Yes, we believe that changes are expected in the in situ XAS even for rods of 200 nm width and 800 nm length. This is because the investigated materials are layer-structured vacancy-containing $\text{Na}_{2x/7}(\text{Mn}_{6/7}\square_{1/7})\text{O}_2$ (\square represents Mn vacancy in the Mn layer). The average distances between the nearest MnO_6 layers in $\text{Na}_x\text{Mn}_3\text{O}_7$ materials are about 3.574-3.691 Å (Supplementary Fig. X1), much larger than the ionic radius of OH^- reactant (1.32 Å). As a result, we believe the corresponding interstitial surface between the MnO_6 layers will be partially available for catalytic reaction, leading to much larger exposed surface than the mere outer surface. Indeed, the variation of in situ XAS spectra was observed in our experiment. Unfortunately, we are unable to obtain the exact volume fraction of the change in $\text{Na}_x\text{Mn}_3\text{O}_7$ due to the substantial contribution from bulk phase in the *in situ* XAS measurement.

Supplementary Fig. X1 | The average layer distance of $\text{Na}_x\text{Mn}_3\text{O}_7$ materials.

For the vacancies in $\text{Na}_x\text{Mn}_3\text{O}_7$ materials, they should be located both in the surface and bulk because a solid-state reaction is adopted using NaNO_3 and MnCO_3 as precursors with controlled Na/Mn ratios. Based on the DFT calculation, OH^- (aq.) tends to spontaneously refill the oxygen vacancy sites of $\text{Na}_x\text{Mn}_3\text{O}_7$ under 1.23 V vs. RHE. Therefore, a small shift of 0.21 eV can be observed in the *in-situ* XAS measurement (Supplementary Fig. 22a), in spite of the substantial complication from the bulk phase.

To better support the claims, as the reviewer suggested, we have moved the *in-situ* Mn 2p and O 1s XPS spectra (Fig. 5a, b) to the main text in the current version. Correspondingly, the *in-situ* XAS spectra (Supplementary Fig. 22) have been moved to the supporting information as an auxiliary explanation.

Comment 1b:

By convention, the x axis of XPS is plotted with the highest energy on the left. The oxidation of the material is an interpretation, not an observation. It is not stated on what observation the interpretation is based. If *in situ* XPS can be performed: why wasn't the O 1s binding studied, which gives more direct insight into oxygen vacancy formation?

Response 1b:

Thanks for the reviewer's comments. We have revised the x axis of XPS with the highest energy on the left. We agree that the oxidation of the material is an interpretation, not an observation, and we have revised the description accordingly. Actually, we have also measured the *in-situ* O 1s XPS spectra of NaMn_3O_7 under open circuit, 1.25 and 1.55 V (versus RHE). From the O 1s XPS spectra (Fig. 5b), the characteristic peak at 531.2 eV corresponding to oxygen vacancy diminishes at an applied potential of 1.25 V compared with that collected at open circuit. Moreover, the characteristic peak at 529.3 eV corresponding to lattice oxygen shifts to a higher energy upon the increase of potential, indicating the oxidation of lattice oxygen. All these indicates that the refilling of oxygen vacancy with OH^- (aq.) and the subsequent deprotonation occurs before the electrochemical OER process.

Change 1b:

1). Page 12, line 21, one paragraph is revised below.

Resolving the near-surface structures under electrochemical condition of the catalyst in its highest metastable catalytic state is a prerequisite for the understanding of the OER mechanism and related active site⁴¹. As such, the *in-situ* X-ray photoelectron spectroscopy (XPS) measurements were performed on NaMn_3O_7 . From Mn 2p XPS spectra (Fig. 5a), the binding energy shifts to a higher energy at an applied potential of 1.25 V compared with that collected at open circuit, indicating the oxidation state of surface Mn increases. With further increase of potential to 1.55 V, no evident variation of Mn 2p spectra indicates that the Mn ions of the catalyst are structurally and electronically similar to that of 1.25 V. From O 1s XPS spectra, the characteristic peak at 531.2 eV corresponding to oxygen vacancy diminishes at an applied potential of 1.25 V compared with that collected at open circuit. Moreover, the characteristic peak at 529.3 eV corresponding to lattice oxygen shifts to a higher energy upon the increase of potential, indicating the

oxidation of lattice oxygen. All these indicates that the refilling of oxygen vacancy with OH^- (aq.) and the subsequent deprotonation occurs before the electrochemical OER process, in agreement with DFT calculations and *in-situ* XAS measurement (Fig. 5c and Supplementary Fig. 21, 22). As shown in Fig. 5c, OH^- (aq.) tends to spontaneously refill the oxygen vacancy sites of $\text{Na}_x\text{Mn}_3\text{O}_7$ under 1.23 V vs. RHE. From Pourbaix diagram (Supplementary Fig. 21), the surface termination of $\text{Na}_x\text{Mn}_3\text{O}_7$ shows the stoichiometric composition under electrochemical OER conditions. Accompanied by the decreased interatomic Mn-O distance from OCV to 1.55 V (Supplementary Fig. 21), the more electrophilic oxygen species with oxygen hole can be formed on the highly covalent oxides after the deprotonation and involve in the subsequent O-O bond formation^{8,28}. Combining all these results, we demonstrate that activating lattice oxygen leads to the enhanced OER activity, as the Fermi level enters the $|\text{O}_{2p}$ states for $\text{Na}_x\text{Mn}_3\text{O}_7$ ($x < 2$) due to the charge compensation and redistribution, creating the reactive oxygen radicals on the surface which behave as electrophilic centers prone to nucleophilic attack from the oxygen lone pairs of OH^- .

2). Page 13, line 10, one figure is revised below.

Fig. 5 | Evidence of lattice oxygen as reaction site under OER conditions. **a,b**, In situ XPS spectra of Mn 2p (**a**) and O 1s (**b**) of NaMn_3O_7 under open circuit, 1.25 and 1.55 V (versus RHE). **c**, The process of oxygen vacancy refilling under 1.23 V and the associated energy barrier. **d**, pH dependence of the OER activities of NaMn_3O_7 . The inset is the proton reaction order estimated by $\rho^{\text{RHE}} = (\partial \log(i) / \partial \text{pH})$, with ρ^{RHE} and i being the proton reaction order and current density at a fixed potential of 1.55 V vs. RHE. **e**, Schematic OER pathway of acid-base nucleophilic attack involving the rate-limiting proton transfer decoupled from electron transfer step. The inset illustrates the formation of negatively charged oxide surface when equilibrated with the electrolyte. **f**, Raman spectra of NaMn_3O_7 electrodes. These electrodes were respectively operated at 1.55 V versus RHE in 1 M KOH (black line) and TMAOH (red line) solution, then

were thoroughly washed by rinsing with high-purity water and acetone for *ex situ* Raman spectroscopy measurement.

Comment 1c:

Fig. 5a does not support the statement “indicating the oxygen vacancy in NaMn_3O_7 can be healed with OH⁻ before the electrochemical OER process”. There is no clear change. Fig. 5b does not support the statement “the amplitude of Mn-O peak increases while the position of Mn-O peak shift towards lower value from OCV to 1.25 V”. The change is within the noise level and not supported by an EXAFS fit. Suppl. Fig. 20 support that there is oxidation from OCV to 1.25 and no further oxidation to 1.55 V. It should be shown in the main text and Fig. 5a and 5b (re)moved. Why should this Mn oxidation be coupled to oxygen vacancy healing and not a deprotonation on the surface, which is commonly proposed?

Response and Change 1c:

Thanks for the reviewer’ constructive comments. We have moved the *in-situ* Mn 2p and O 1s XPS spectra to the main text. Correspondingly, the *in-situ* XAS spectra have been moved to the supporting information as an auxiliary explanation. From Mn 2p XPS spectra (Fig. 5a), the binding energy shifts to a higher energy at an applied potential of 1.25 V compared with that collected at open circuit, indicating the oxidation state of surface Mn increases. With further increase of potential to 1.55 V, no evident variation of Mn 2p spectra indicates that the Mn ions of the catalyst are structurally and electronically similar to that of 1.25 V. From O 1s XPS spectra, the characteristic peak at 531.2 eV corresponding to oxygen vacancy diminishes at an applied potential of 1.25 V compared with that collected at open circuit. Moreover, the characteristic peak at 529.3 eV corresponding to lattice oxygen shifts to a higher energy upon the increase of potential, indicating the oxidation of lattice oxygen. All these indicates that the refilling of oxygen vacancy with OH⁻ (aq.) and the subsequent deprotonation occurs before the electrochemical OER process. More information can be found in **Response 1b and Change 1b**.

Comment 2:

The replies to the reviewers are clear. Please add it to the manuscript or Supplemental information so that the readers can follow your arguments.

Response 2:

Thanks for the reviewer’ valuable suggestion. We have added the replies to the Supplemental information so that the readers can follow the arguments.

Change 2:

1). Supporting information, the significance of the magnetization moments of the oxygen ions has been added.

Supplementary Table 1 | Computed magnetization moment and charge of oxygen ions in $\text{Na}_x\text{Mn}_3\text{O}_7$. Magnetization moment results from the presence of unpaired electrons in materials, and the fully paired electrons lead to a net magnetic moment of zero. For $\text{Na}_2\text{Mn}_3\text{O}_7$, the oxygen ions tend to form 8-electron stable structures with no unpaired electrons. When the number of Na^+ decreases from $x=2$, lattice oxygen

ions are activated and oxidized, meanwhile Mn would be in their maximum achievable oxidation states of 4+. Thus, the unpaired electrons are generated in such oxidized oxygen ions.

Model	Magnetization moment (μB)	Charge (e)
Na ₂ Mn ₃ O ₇ -S1	0.11	-1.03
Na _{1.5} Mn ₃ O ₇ -S2	0.17	-1.00
Na _{1.5} Mn ₃ O ₇ -S3	0.23	-0.94
NaMn ₃ O ₇ -S4	0.30	-0.89
Na _{0.5} Mn ₃ O ₇ -S5	0.34	-0.80
Na _{0.5} Mn ₃ O ₇ -S6	0.60	-0.66

Comment 3:

Thanks for performing EXAFS simulations. Please perform them with identical σ^2 (in addition to the free fits) to support reduction of the coordination numbers CN. CN and sigma2 correlate strongly. It is suspicious that both fit variables are reduced. Shouldn't R increase linearly with the reducing the Na stoichiometry?

Response 3:

We have fitted Mn K-edge EXAFS curves with identical σ^2 according to the suggestion from the reviewer. The corresponding results are showed in Supplementary Table 7. As the number of Na⁺ is reduced, the coordination number of Mn-O decreases while the interatomic distance of Mn-O increases.

Sample	bond	^a R (Å)	^b CN	^c σ^2 (Å ²)	^d R factor
Na ₂ Mn ₃ O ₇	Mn-O	1.907	6.0 (fixed)	0.005	0.016
Na _{1.5} Mn ₃ O ₇	Mn-O	1.910	5.78	0.005	0.012
NaMn ₃ O ₇	Mn-O	1.914	5.31	0.005	0.023
Na _{0.7} Mn ₃ O ₇	Mn-O	1.918	4.86	0.005	0.032

Comment 4:

Fig. 3e does not show the O-K main edge. No conclusions can be made about it. The pre-edge is shown. Please update the text accordingly. Please discuss for the readers why the shift of the pre-edge onset is fully assigned to the oxygen valence and not to changes of the Mn orbitals as is discussed on P10, L9. Are there other reports in literature where oxygen holes are studied by the O K pre-edge? Nat. Commun. 2020, 11, 4973 - cannot find it. A typo? Nat. Mater. 2019, 18, 256) -> clearly states "O K-edge sXAS and RIXS further support pure multivalent, hybridized Ir-O redox in the absence of oxygen redox in LIO." I find the interpretation of the O-K pre-edge is still insufficiently discussed or unsupported by the data.

Response 4:

We really appreciate the professional comment from the reviewer. In the past, the variations in either the intensity or the lineshape in the O K pre-edge in battery electrodes were considered as an indicator of oxidized oxygen or oxygen hole states. By literature review, we strongly believe that the variation of O K pre-edge cannot simply be linked to the change of oxygen valence. (*Nat. Chem.* **2016**, *8*, 684; *Nat. Commun.* **2016**, *7*, 13814; *Nat. Chem.* **2018**, *10*, 288). Recently, using experiment and theoretical calculations, Yang et al. demonstrated that the O-K pre-edge lineshape is dominated by the type of metal and their oxidation states, because the low-lying unoccupied 3d states of different metal could all be projected onto this common energy window around 525-535 eV (*Energy Environ. Mater.* **2020**, *0*, 1). As a result, a higher Mn oxidation state gives rise to a lower O-K pre-edge peak, which is exactly the opposite to the typical behavior of the Mn L-edge that display higher energy for higher oxidation states. As expected in Fig. 3e, the evident shift of the O K pre-edge to a higher energy region can be observed. It indicates the decrease of Mn oxidation state owing to the increase of oxygen vacancy when the number of Na⁺ decreases, in agreement with the Mn L-edge XAS spectra (Fig. 3e).

Change 4:

1). Page 10, line 11, three sentences are revised below.

Moreover, the evident shift of the O K pre-edge to a higher energy region further confirms the decrease of Mn oxidation state owing to the increase of oxygen vacancy when the number of Na⁺ decreases, in agreement with the Mn L-edge XAS spectra (Fig. 3e). As mentioned later, the OH(aq.) tends to spontaneously fill the oxygen vacancy sites of NaMn₃O₇ under electrochemical OER conditions (Fig. 5c). On further electrochemical deprotonation, the lattice oxygen ions coordinated with two Mn ions would begin to be oxidized, producing oxygen hole states in |O_{2p}.

2). Page 9, line 1, one figure is revised below.

Fig. 3 | Electronic and atomic coordination structures of $\text{Na}_x\text{Mn}_3\text{O}_7$. **a**, XRD pattern of $\text{Na}_2\text{Mn}_3\text{O}_7$. **b,c**, Normalized Mn K-edge XANES spectra (**b**) and EXAFS oscillation functions (**c**) of $\text{Na}_x\text{Mn}_3\text{O}_7$ ($x = 2.0, 1.5, 1.0$ and 0.7). **d**, Fourier transform magnitudes of Mn K-edge EXAFS $k^2\chi(k)$. **e,f**, Mn L_{2,3}-edge (**e**) and **O K pre-edge** (**f**) of soft XAS spectra of $\text{Na}_2\text{Mn}_3\text{O}_7$ and NaMn_3O_7 .

Comment 5:

Please add the definition of “lattice oxygen” to the figure or its caption.

Response 5:

Thanks for the reviewer’s comment. We have added the definition of “lattice oxygen” to the figure and its caption.

Change 5:

1). Page 5, line 1, one figure is revised below.

Fig. 1 | Activating lattice oxygen to regulate the pathway competition for O-O bond formation. a, Schematic formation of oxygen holes in $|O_{2p}$ lone pair states for $Na_xMn_3O_7$. Lattice oxygen atoms are defined as O1 and O2, where O1 is coordinated with three Mn ions and O2 is coordinated with two Mn ions, respectively. **b,** Projected density of states of $Na_xMn_3O_7$ slabs ($x = 2, 1.5, 1$ and 0.5). **c,** Partial charge density projected on O atoms by the shaded region shown in Fig. 1b. **d,** Free energy difference between the two isomeric intermediates of A3 (*O) and R4 (*OO*) on representative oxygen coordination environments. The details about A3 and R4 are shown in Supplementary Fig. 4. **e,** Activation free energy barrier (ΔG^\ddagger) from A3 to R4 on the specific coordination environment of S6. **f,** Free energy barrier (ΔG^{*OOH}) for the formation of *OOH.

Comment 6:

Fig. SS3 should be added as supplemental figure as it is important to understand the connection between P1 and P2 as well as the limits for materials optimization using Na stoichiometry.

Response 6:

We appreciate the reviewer's valuable suggestion. It is true that Fig. SS3 is helpful to understand the connection between P1 and P2 as well as the limits for materials optimization using Na stoichiometry. Thus, we have added it as Supplementary Fig. 8 in this revised version.

Supplementary Fig. 8 | Correlation between the parameters of P1 and P2 and the number of Na^+ in $Na_xMn_3O_7$.

Comment 7:

The documentation of the XAS experiments is still insufficient. What detector was used for the fluorescence yield? How was the synchrotron beam monochromatized? What beamline was used? Details of the Fourier transform are missing. The work is not reproducible when the energy/k-range and window function of the Fourier transform is omitted.

Response 7: We appreciate the reviewer's comments. According to the suggestion, we have added the missing information or details about the XAS experiments. XAS experiments were performed at the X-ray Absorption Fine structure for catalysis (XAFCA) beamline at the Singapore Synchrotron Light Source (SSLS). The fluorescence yields from the sample can be measured using a silicon drift detector (Bruker Xflash 6|100), which has a 100 mm² active area in a single element and a 600 Kcounts per second pulse loading capability. XAFCA covers the photon energy range from 1.2 to 12.8 keV, making use of two sets of monochromator crystals, an Si (111) crystal for the range from 2.1 to 12.8 keV and a KTiOPO₄ crystal [KTP (011)] for the range between 1.2 and 2.8 keV. The k^2 -weighted Fourier transforms for all of the EXAFS data were conducted in the k -range 2-12 Å⁻¹ using a Hanning-shaped window. The k - and R -ranges for the fitting of all of the EXAFS data were set as 2-12 Å⁻¹ and 1.0-3.5 Å, respectively.

Change 7:

1). Supporting information, the details about XAS experiments are added as follows.

The fluorescence yields from the sample were measured using a silicon drift detector (Bruker Xflash 6|100), which has a 100 mm² active area in a single element and a 600 Kcounts per second pulse loading capability. XAFCA covers the photon energy range from 1.2 to 12.8 keV, making use of two sets of monochromator crystals, an Si (111) crystal for the range from 2.1 to 12.8 keV and a KTiOPO₄ crystal [KTP (011)] for the range between 1.2 and 2.8 keV. The k^2 -weighted Fourier transforms for all of the EXAFS data were conducted in the k -range 2-12 Å⁻¹ using a Hanning-shaped window. In situ XAS measurements were performed in fluorescence mode using a home-made electrochemical cell. The catalysts were sprayed on carbon paper at a loading of 2 mg cm⁻² as the working electrode. Acquired extended XAFS (EXAFS) data were processed according to standard procedures using the ATHENA module implemented in the IFEFFIT software packages⁵. The k - and R -ranges for the fitting of all of the EXAFS data were set as 2-12 Å⁻¹ and 1.0-3.5 Å, respectively.

REVIEWERS' COMMENTS

Reviewer #3 (Remarks to the Author):

Huang et al have further revised their manuscript to clarify the remaining questions. All concerns have been addressed well. Highlighting the in situ XPS data significantly strengthened the manuscript. I now fully recommend publication in Nature Communications.